

# The next generation sea-ice model neXtSIM, version 2

Einar Ólason[1,2], Guillaume Boutin[1,2], Timothy Williams[1,2], Anton Korosov[1,2], Heather Regan[1,2], Jonathan Rheinlænder[1,2], Pierre Rampal[3], Daniela Flocco[4], Abdoulaye Samaké[5], Richard Davy[1,2], Timothy Spain[1,2], and Sean Chua[1,2]

[1]Nansen Environmental and Remote Sensing Center
[2]Bjerknes Centre for Climate Research, Bergen, Norway
[3]CNRS, Institut des Géosciences de l'Environnement, Grenoble, France
[4]Dipartimento di Scienze della Terra, dell'Ambiente e delle Risorse, Università degli Studi di Napoli "Federico II", Napoli, Italy
[5]Université des Sciences, des Techniques et des Technologies de Bamako, Bamako, Mali

**Correspondence:** Einar Ólason (einar.olason@nersc.no)

**Abstract.** While many large-scale sea ice models can represent regional to global sea ice evolution, their representation of sea ice dynamics varies little between models. This is because they all use rheologies based on the hypothesis that sea ice behaves as a visco-plastic solid. This works reasonably well for several quantities (e.g. sea ice volume) but fails to capture sea ice deformation features at coarse and moderately high resolutions (i.e. coarser than about 5 km resolution). This may
be problematic since these deformations result in the formation of leads and ridges, which likely play an essential role in ice-atmosphere-ocean interactions, and because these are the resolutions at which sea ice models run in coupled models such as Earth System Models. An alternative is to use brittle rheologies that better capture these features independently of the resolution. The neXtSIM model has been at the core of the effort by its developers and users to explore the usage of brittle rheologies and new modelling approaches in geophysical scale simulations of sea ice. Here, we document neXtSIM, now in
version 2 of its development, to foster its use for the sea ice community and release a public version of the model. We describe the sea ice dynamics and the core of the model in detail and give insights into the parameters specific to the brittle rheologies included in neXtSIM. We also document the model's specificity associated with its Lagrangian framework and how it affects the coupling with other components of Earth system models. We hope that the insights provided in this study and the public release of the model will trigger innovative research in the sea ice modelling community.

## 15  1  Introduction

The next-generation sea-ice model, neXtSIM, has been developed at the Nansen Center in Bergen since 2012 at the instigation of P. Rampal and his group. It was conceived as an opportunity to create a new modelling tool for research, offering some unique features, including i) a continuous and fully Lagrangian framework and ii) a new treatment of sea ice dynamics through a different class of rheology inspired by damage mechanics. The dynamical core of neXtSIM was introduced by Bouillon and
Rampal (2015). This was based on the pioneering work by Girard et al. (2011), who introduced the first brittle rheology to be applied to sea ice (the elasto-brittle or EB). Numerical modifications to the initial implementation of Girard et al. (2011)



allowed Bouillon and Rampal (2015) to run neXtSIM for ten days, while Girard et al. (2011) only ran the model for four days (including one-day spin-up) and assumed a quasi-static regime (i.e. left-end side term of equation 1 in section 2 equal to 0) in their initial work. Thanks to the EB rheology, the new model could reproduce the multifractal spatial scaling properties of
sea-ice deformation. Using a Lagrangian advection scheme, the model could preserve the resulting discontinuous fields well.

Further development of neXtSIM was focused on turning the dynamical core of Bouillon and Rampal (2015) into a fully-fledged sea-ice model. Rampal et al. (2016a) presented a significant step towards this goal with the addition of thermodynamic melt and growth of the ice and dynamical remeshing of the model's triangular mesh. This allowed Rampal et al. (2016a) to run the model for an entire year, showing realistic ice thickness and ice extent evolution while reproducing the multifractal spatial
scaling of sea-ice deformation, as before. Following these developments, the model was ported from the original Matlab/C hybrid code to a C++ code base and parallelised using the distributed-memory approach with the MPI message-passing library (Samaké et al., 2017).

The Maxwell elasto-brittle rheology (MEB Dansereau et al., 2016) was implemented into neXtSIM following its publication, and Rampal et al. (2019) used this latest version of the model to compare for the first time the simulated temporal, as well as the
spatial scaling of sea-ice deformation with the same type of scaling relationships found in the observations Marsan et al. (2004); Rampal et al. (2008). The brittle Bingham-Maxwell rheology was developed and implemented in neXtSIM, allowing Ólason et al. (2022) to run the model over several years, showing realistic ice thickness and ice extent evolution while reproducing the multifractal spatial scaling of sea-ice deformation and realistic pan-Arctic deformation patterns. Boutin et al. (2023) extended this work, using neXtSIM coupled to the ocean model of NEMO to explore the mass balance of Arctic sea ice over several
decades using BBM. These latest developments have allowed the neXtSIM development team to reach their goal of turning the model into a fully-fledged sea-ice model, albeit with some caveats.

This paper aims to provide an overview of the features of neXtSIM that have already been documented and to document those that have remained undocumented. The model description is, by its nature, technical, but we will strive to highlight the physical reasoning behind the modelling choices we have made. A core tenet of the neXtSIM development process is to use
the simplest modelling approach, which can reproduce observations, of which satellite remote sensing observations are most important.

## 2   Dynamics

The core equation of sea-ice dynamics is the momentum equation. The form used in neXtSIM is

$$m\frac{\partial \boldsymbol{u}}{\partial t} = \boldsymbol{\nabla} \cdot (\boldsymbol{\sigma} h) + A(\boldsymbol{\tau}_a + \boldsymbol{\tau}_w) + \boldsymbol{\tau}_b + mf\boldsymbol{k} \times \boldsymbol{u} - mg\boldsymbol{\nabla}\eta, \tag{1}$$

where $m = A\rho h$ is the ice mass per unit area, $\boldsymbol{u}$ is the ice velocity, $\boldsymbol{\sigma}$ is the internal stress tensor, $h$ is the ice slab thickness (not volume per unit area), $\rho$ the ice density, $\boldsymbol{\tau}_a$ and $\boldsymbol{\tau}_w$ are the atmosphere and ocean stress terms, respectively, $\boldsymbol{\tau}_b = -C_b\boldsymbol{u}$ is the basal stress term introduced in Lemieux et al. (2015), $mf\boldsymbol{k} \times \boldsymbol{u}$ is the Coriolis term, with vertical unit vector $\boldsymbol{k}$, and $mg\boldsymbol{\nabla}\eta$ is the ocean-tilt term. We write explicitly the integrated internal stress as $\boldsymbol{\sigma} h$ following Sulsky et al. (2007) and Bouillon and Rampal (2015). The rheology determines the internal stress, $\boldsymbol{\sigma}$, which links stress and strain or strain rates in the ice.





One primary purpose of neXtSIM has been to explore the performance and effects of using the brittle rheologies (Girard et al., 2011; Dansereau et al., 2016; Ólason et al., 2022) in a large-scale context. In version 2 of neXtSIM, only the BBM rheology is included, out of the brittle rheologies. The EB and MEB rheologies can be recovered using a suitable set of parameters for BBM. Still, the interest in using EB and MEB for large-scale applications is, arguably, limited.

In version 2 of neXtSIM, we also included an implementation of the mEVP rheology (Bouillon et al., 2013). This was initially an EVP implementation as a technical exercise to aid the BBM development. This was then modified to an mEVP to act as a benchmark to compare BBM against (Ólason et al., 2022).

## 2.1 BBM implementation

The BBM rheology is described in detail in Ólason et al. (2022), but we will discuss it briefly here for completeness, focusing on any neXtSIM-related specifics.

The BBM rheology is a damaging Bingham-Maxwell constitutive model. Using damage mechanics for sea-ice modelling was introduced with the EB model by Girard et al. (2011), but damage mechanics are widely used in other communities, e.g. in rock and crustal mechanics (e.g. Lyakhovsky et al., 1997; Amitrano et al., 1999; Schapery, 1999)). The EB model simulates the ice as a damaging elastic sheet, where each grid element can be considered a spring in a mechanics sense. In it, the stress is calculated in every grid cell, and if the stress is outside the Mohr-Coulomb yield criterion, the elasticity is reduced in that grid cell, and the sea ice experiences a reversible deformation. Elasticity in the model is thus

$$E = Y(1-d), \tag{2}$$

where $Y$ is the ice's Young modulus, and $d \in [0, 1[$ is a scalar damage variable. At the start of the simulation, $d = 0$ everywhere, but $d$ is then reduced to ensure that all stresses in the ice are always within the yield criterion. A local increase in $d$ causes a stress redistribution and a cascade of damage reduction emulating the multiplicative cascade Marsan et al. (2004) suggested was the reason for the observed spatial scaling of sea-ice deformation they observed.

The MEB model (Dansereau et al., 2016) introduced a viscous element, a dashpot, in series with the elastic one. This is intended to simulate the viscous dissipation of internal stresses and the larger - irreversible - deformations (as opposed to the elastic counterpart) occurring along faults once the material is highly damaged. The viscosity of the dashpot is very high when the ice is undamaged but should decrease faster than the spring's elasticity when damage increases. This is formulated as:

$$\eta = \eta_0 (1-d)^\alpha, \tag{3}$$

where $\alpha$ is a material-dependent constant.

The Bingham-Maxwell model consists of a friction block and a dashpot in parallel, connected with a spring in serial. A schematic of the constitutive model is shown in panel a of figure 1. The spring's elasticity and the dashpot's viscosity evolve as a function of the damage variable, as in the MEB rheology. The friction element is introduced to emulate the resistance to ridging so that for small compressive stresses, the ice remains fully elastic, regardless of the level of damage. This leads to the model having three regimes in stress space: a visco-elastic converging one for high compressive normal stress, a purely elastic





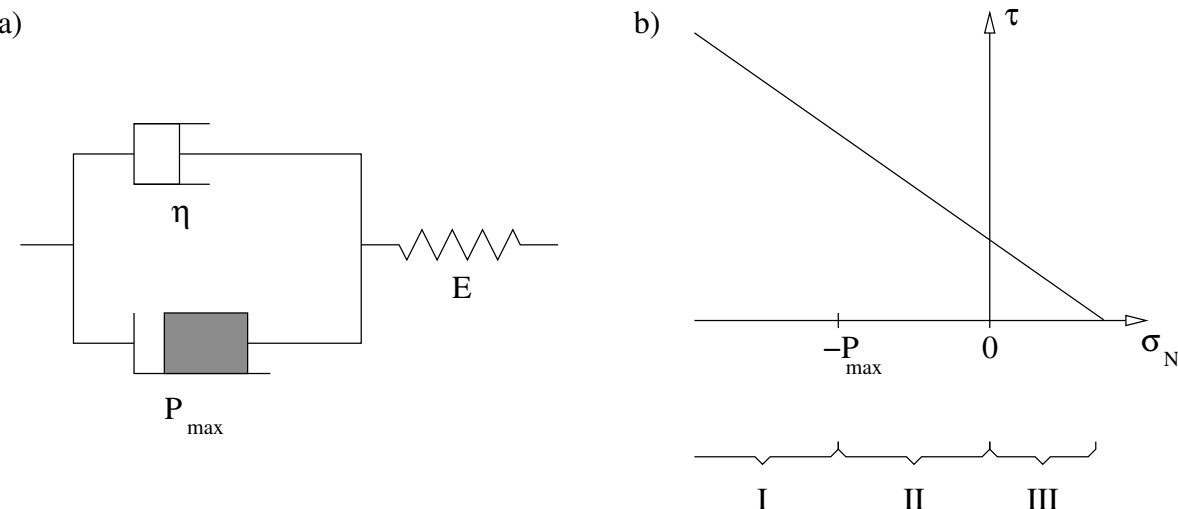

**Figure 1.** Panel a) A schematic of the Bingham-Maxwell constitutive model showing a dashpot and a friction element connected in parallel, with both connected to a spring in series. Panel b) The yield criterion in the stress invariant plane $\{\sigma_N, \tau\}$, as well as the elastic limit $P_{\mathrm{max}}$, and the ridging (I), elastic (II), and diverging (III) regimes.

one for compressive normal stress less than the threshold $P_{\mathrm{max}}$, and a visco-elastic diverging one for divergent normal stress. These three regimes are shown schematically in panel b of figure 1.

The constitutive equation for BBM is

$$\dot{\sigma} = E\mathbf{K} : \dot{\varepsilon} - \frac{\sigma}{\lambda}\left(1 + \widetilde{P} + \frac{\lambda \dot{d}}{1-d}\right), \tag{4}$$

where $\sigma$ and $\dot{\varepsilon}$ are the stress and strain rate tensors, respectively, $\lambda = \eta/E = \lambda_0(1-d)^{\alpha-1}$ is the viscous relaxation time, with $\lambda_0$ the undamaged viscous relaxation time. The time derivative is noted with a dot. The resistance to ridging is encoded in $\widetilde{P}$ as

$$\widetilde{P} = \begin{cases} \frac{P_{\mathrm{max}}}{\sigma_N} & \text{for } \sigma_N < -P_{\mathrm{max}} \\ -1 & \text{for } -P_{\mathrm{max}} < \sigma_N < 0 \\ 0 & \text{for } \sigma_N > 0 \end{cases} \tag{5}$$

where

$$P_{\mathrm{max}} = P\left(\frac{h}{h_0}\right)^f e^{-C(1-A)}, \tag{6}$$

with $h_0 = 1$ m, $A$ is ice concentration and $C$ is a constant regulating the decrease of $P_{\mathrm{max}}$ as the concentration decreases. The factor $f \in [1, 2]$ was set to $f = 3/2$ by Ólason et al. (2022), who also briefly explored the impact of varying $f$. The MEB model may be recovered by setting $\tilde{P} = 0$, while the EB model is recovered when $\tilde{P} = -1$.



The time stepping of equation (4) is done via a time-splitting method, where multiple dynamic time steps are taken (of order 100) within a model time step. The time stepping is done by first calculating an intermediate stress

$$\sigma' = \frac{\lambda(\Delta t E\mathbf{K} : \dot{\varepsilon} + \sigma^n)}{\lambda + \Delta t(1 + \widetilde{P})}. \tag{7}$$

If this stress is outside the yield criterion, an updated damage value is calculated as

$$d^{n+1} = d^n + (1 - d_{\mathrm{crit}})(1 - d^n), \tag{8}$$

where

$$d_{\mathrm{crit}} = \begin{cases} -N/\sigma_N & \text{if } \sigma_N < -N, \\ c/(\tau + \mu\sigma_N) & \text{if } \tau > c - \mu\sigma_N, \end{cases} \tag{9}$$

and where $\sigma_N$ is the mean normal stress, $\tau$ is the maximum shear stress, and $c$ and $\mu$ are the cohesion and internal friction coefficient values for the Mohr-Coulomb criterion, respectively. A capping with a large value of $N$ is needed to prevent numerical instabilities (see Plante et al., 2020). Damage (and stress) are only updated if the stress is outside the yield criterion, i.e. $0 < d_{\mathrm{crit}} < 1$.

Following the damage update, a consistent stress correction is applied

$$\sigma^{n+1} = \sigma' - (1 - d_{\mathrm{crit}})\sigma'. \tag{10}$$

Note that equations (8) and (10) differ from those of Ólason et al. (2022) by a factor of $\Delta t/t_d$ in the second term on the right, where $\Delta t$ is the dynamic time step and $t_d$ is a damage propagation time scale. Doing so, we follow the damage and stress relaxation scheme of Bouillon and Rampal (2015) rather than that of Dansereau et al. (2016), as Ólason et al. (2022) did. The consequences of this are discussed in section 5.1.

The strain rates needed for the stress calculation are spatial derivatives of the velocity, calculated using a spatial discretisation of the equations. The spatial discretisation of neXtSIM is unusual among large-scale sea-ice models and is particularly advantageous for the BBM implementation. NeXtSIM uses a finite-element discretisation, where tracers are constant on the elements (P0), while the velocities are located at the nodes of the mesh triangles (P1). This configuration means the elements' strain rate and stress tensors are constant, with all their components co-located. This co-location is advantageous because all stress tensor components are needed to calculate $\sigma_N$ and $\tau$ of equation (9). Models using a staggering where different components of the stress tensor lie at different locations in the grid (e.g. the C-grid) must apply extra effort to achieve the co-location of the stress components needed to get an accurate estimate of $d_{\mathrm{crit}}$ (e.g. Plante et al., 2020; Brodeau et al., 2024).

## 2.2 mEVP implementation

The mEVP rheology (Lemieux et al., 2012; Bouillon et al., 2013) is a numerically efficient and easily parallelisable approach to solving the original VP equations (Hibler, 1979). In the VP model, the internal stress tensor is diagnosed from the strain




rates as

$$\sigma_N = \zeta\dot\varepsilon_I - P/2 \tag{11}$$

$$\tau = \eta\dot\varepsilon_{II} \tag{12}$$

where $\dot\varepsilon_I$ and $\dot\varepsilon_{II}$ are the first and second strain rate invariants, and

$$\zeta = \frac{P}{2\Delta} \tag{13}$$

$$\eta = \zeta/e^2 \tag{14}$$

$$\Delta = \sqrt{\dot\varepsilon_I^2 + \dot\varepsilon_{II}^2/e^2} \tag{15}$$

$$P = P^* h e^{-C(1-A)} \tag{16}$$

with $e$ and $P^*$ constants.

The VP equations must be solved implicitly with an iteration scheme such as a fixed-point iteration or Picard solver. To avoid using an implicit solver, Hunke and Dukowicz (1997) introduced an elastic term, which allows the equations to be solved explicitly, using sub-iterations. The VP rheology is recovered as the steady state limit of the EVP rheology of Hunke and Dukowicz (1997) or in the limit of an infinitely large elasticity. For the EVP, the momentum equation is solved using an explicit time-stepping scheme identical to that used for the BBM.

The mEVP, in addition to modifying the constituent model, also modifies the momentum equation. In neXtSIM, we take advantage of similarities between the EVP/BBM momentum solver and the mEVP one. For mEVP, we can rewrite the modified momentum equation of mEVP (equations (39) and (40) of Lemieux et al., 2012) can be written as

$$\beta'(\mathbf{u}^{n+1} - \mathbf{u}^n) = \mathbf{u}^0 - \mathbf{u}^{n+1} - \Delta t \mathbf{f} \times \mathbf{u}^{n+1}$$

$$+ \frac{\Delta t}{m}\left[\mathbf{F} + AC_d\rho_w(\mathbf{u}_w - \mathbf{u}^{n+1})|\mathbf{u}_w - \mathbf{u}^{n+1}| - mg\nabla\eta\right], \tag{17}$$

where $\beta'$ is the numerical tuning parameter introduced by Lemieux et al. (2012), $\mathbf{u}^0$ is the ice velocity at the start of the sub iterations, $\mathbf{u}^n$ and $\mathbf{u}^{n+1}$ is the velocity at previous and current sub-iteration step, and $\mathbf{F}$ donates all terms not dependent on, or multiplied with $\mathbf{u}$. Using the notation and approach of Hunke and Dukowicz (1997), this can be rewritten as

$$(\alpha^2 + \beta^2)u_1^{n+1} = \alpha u_1^n + \beta u_1^n + \frac{u_1^0 - u_1^n}{\beta' + 1}$$
$$+ \frac{\Delta t}{(\beta'+1)m}\left[\alpha\left(\frac{\partial\sigma_{1j}^{n+1}}{\partial x_j} + \tau_1\right) + \beta\left(\frac{\partial\sigma_{2j}^{n+1}}{\partial x_j} + \tau_2\right)\right] \tag{18a}$$

$$(\alpha^2 + \beta^2)u_2^{n+1} = \alpha u_1^n + \beta u_2^n + \frac{u_2^0 - u_2^n}{\beta' + 1}$$
$$+ \frac{\Delta t}{(\beta'+1)m}\left[\alpha\left(\frac{\partial\sigma_{2j}^{n+1}}{\partial x_j} + \tau_2\right) + \beta\left(\frac{\partial\sigma_{1j}^{n+1}}{\partial x_j} + \tau_1\right)\right] \tag{18b}$$




where $\alpha$, $\beta$, and $\tau_{\{1,2\}}$ are as defined in Hunke and Dukowicz (1997). Importantly, the BBM and EVP equations are recovered by setting $\beta' = -1$ and $\mathbf{u}^0 = \mathbf{u}^n$.

In neXtSIM, switching between the BBM and mEVP rheologies is done by calling separate functions to calculate the stress tensor, $\sigma$. The momentum equation is then solved using equations (18a) and (18b), setting $\beta' = -1$ and $u^0 = u^n$ when using BBM.

## 3   Thermodynamics and column physics

The thermodynamics and column physics in neXtSIM are relatively simple compared to other sea-ice models, focusing on
representing the processes most likely to be of interest when using brittle rheology, e.g. atmosphere–ocean–ice interactions in leads and polynyas. In addition, a diagnostic ridge ratio scheme is implemented in the model for the benefit of the sea-ice forecasting platform, neXtSIM-F (Williams et al., 2021) and a melt-pond scheme to improve long-term simulations, such as those of Boutin et al. (2023) and Regan et al. (2023). An ice-age tracer was implemented for Regan et al. (2023), with subsequent minor improvements.

### 165  3.1   Column thermodynamics implementation

A sea-ice model's grid cell must be divided into at least two categories: ice and open water (Hibler, 1979, e.g.). Many models also use categories of different ice thicknesses (Hibler, 1980, e.g.), with five thickness categories being a common choice (Hunke, 2001, particularly after). NeXtSIM can use the minimal two categories of ice and open water or a three-category approach, categorising ice as either young or consolidated (see appendix A2 of Rampal et al., 2019, who refer to the categories
as "thick" and "thin" and not "consolidated" and "young"). Vertical ice growth in the two ice classes is calculated by solving the heat diffusion equation in one dimension, following the zero-layer approach of Semtner (1976) for the young-ice class and the two-layer approach of Winton (2000) for the consolidated ice class.

    Ice growth in open water occurs when the ocean surface becomes supercooled. This ice is either added to the one ice class using Hibler's (1979) collection-depth approach or to the young-ice class following Rampal et al. (2019). In melting conditions,
the ice concentration should decrease due to lateral melt. Young ice melts first before the consolidated ice concentration decreases when using the young-ice class. NeXtSIM uses the lateral melt scheme of Mellor and Kantha (1989), which uses a fraction of the ice-ocean heat flux to melt ice laterally while the rest warms the ocean mixed-layer.

    The two-layer approach of Winton (2000) discretises the heat-diffusion equation using two internal temperature points in the ice and assumes zero heat capacity in the snow. It also accounts for the change in the internal enthalpy in the ice due
to the presence of brine. Although substantially simpler than the more complete schemes (e.g. Bitz and Lipscomb, 1999; Turner and Hunke, 2015), this approach can capture the seasonal cycle and the main characteristics of thermodynamic melt and growth. The zero-layer approach of Semtner (1976) assumes that the ice has no internal temperature and no heat capacity. This straightforward approach is insufficient for general use (Semtner, 1984), but suffices to reproduce the melt and growth of





ice in the young ice class, as this is generally thinner than about 30 cm. Therefore, neXtSIM uses the zero-layer approach for
ice in the young-ice class, as it is both simpler and faster in execution. It uses the two-layer approach for the consolidated ice.

For both two- and zero-layer models, it is assumed that the ice temperature at the base is at the freezing point of the seawater
below. The surface temperature is calculated by balancing the heat flux through the ice with the fluxes from the surface into the
atmosphere as

$$Q + \frac{\partial Q}{\partial T_s} \Delta T_s = \frac{k_i k_s}{k_s h_i + k_i h_s} \left( T_s + \Delta T_s - T_i \right). \tag{19}$$

Here $Q$ is the sum of the latent, sensible, short-wave, and long-wave fluxes at the surface, $T_s$ is the surface temperature at the
previous time step, $\Delta T_s$ is the change in surface temperature, $k_s$ and $k_i$ are the snow and ice heat conductivities, respectively,
and $h_s$ is the snow thickness. $T_i$ is the ice temperature at the upper-temperature point, $T_1$ in the case of the two-layer model,
or the temperature of the ice base in the case of the zero-layer model. For the two-layer model, $h_i$ is the thickness of ice above
the $T_1$ temperature point, while for the zero-layer model, $h_i$ is the total ice thickness.

The reanalysis prescribes the incoming short-wave and long-wave fluxes, while the model calculates the albedo, short-wave
penetration, and outgoing long-wave radiation. The albedo is calculated using a simplified meltpond scheme (see section 3.2),
and the outgoing long-wave radiation and its derivative are simply the black-body radiation and its derivative.

The turbulent fluxes are calculated using bulk formulas relating the near-surface atmospheric and ice surface states to surface
fluxes, as neXtSIM has so far always been forced by atmospheric model results. The bulk formulas for sensible and latent heat
fluxes are:

$$Q_s = C_h \rho_a c_p |\mathbf{u}_a| (T_s - \theta) \tag{20}$$

$$Q_l = C_h \rho_a L_s |\mathbf{u}_a| (q_s - q_a), \tag{21}$$

where $C_h$ is a drag coefficient, $\rho_a$ atmospheric density, $c_p$ the specific heat capacity of air, $L_s$ the latent heat of sublimation,
$|\mathbf{u}_a|$ is the wind speed, $\theta$ the potential temperature at the reference height, $q_s$ the specific humidity at the surface, and $q_a$ the
specific humidity at the reference height. The specific humidity may be supplied by the reanalysis or calculated from the mixing
ratio or dew-point temperature.

The drag coefficients $C_h$ and $C_m$ used to calculate wind drag do not consider surface roughness; they only take atmospheric
stability. The drag coefficients then become

$$C = \frac{k^2}{\left[ \ln\left( \frac{z}{z_0} \right) - \Psi(\zeta) \right]^2}, \tag{22}$$

where $k$ is the von Kármán constant, $z$ is the reference height, $z_0$ the surface roughness length, $\Psi$ the stability function, and
$\zeta = z/L$ the stability parameter, where $L$ is the Obukov length. We use the stability functions of Grachev et al. (2007) and
Kader and Yaglom (1990) for both momentum and heat and calculate separate drag coefficients and stability functions for
momentum and heat, using different reference heights (10 m and 2 m by default, respectively). The inverse of the Obukov
length

$$\frac{1}{L} = -\frac{k g (\overline{w' \theta_v'})_s}{u_*^3 \bar{\theta}_v} \tag{23}$$





is calculated by approximating some key parameters. The surface virtual potential temperature flux is approximated as

$$\overline{(w'\theta_v')}_s = \overline{w'\theta'}(1 + ar) + a\theta\overline{w'r'}, \tag{24}$$

where the potential temperature flux is approximated as

$$\overline{w'\theta'} = \hat{C}_h|\mathbf{u}|(T_s - \theta), \tag{25}$$

the mixing ratio flux as

$$\overline{w'r'} = \hat{C}_h|\mathbf{u}|\frac{q_s - q_a}{(1 - q_s)(1 - q_a)}, \tag{26}$$

and where $a = 0.6078$, $r = q_a/(1 - q_a)$ is the mixing ratio, $\bar{\theta}_v = \theta(1 + aq_a)$ is the virtual potential temperature, and $\theta = T + \Gamma_d z_h$ is the potential temperature, with $\Gamma_d$ being the adiabatic lapse rate. The frictional velocity is approximated as

$$u_* = \sqrt{\hat{C}_m}|\mathbf{u}|. \tag{27}$$

$\hat{C}_m$ and $\hat{C}_h$ are the drag coefficient of the previous time step. The inverse Obukov length is limited to $1/L \in [-1, 1]$ m$^{-1}$. As advecting additional prognostic variables is very cheap, we set the drag coefficients as prognostic variables and forego iterating over these equations using the neutral drag coefficients at the start.

## 3.2 Melt pond scheme

The neXtSIM model includes two albedo parameterisations that account for the evolution of melt ponds. The first one is the same as the "CCSM3" albedo scheme (Briegleb et al., 2004). The CCSM3 scheme is a shortwave radiation scheme that includes a melt pond parameterisation, representing the melt pond effect by reducing the albedo value when the surface temperature of sea ice increases. The second one is a recent addition and explicitly represents melt ponds.

The neXtSIM melt-pond scheme largely follows the work of Flocco et al. (2010), Holland et al. (2012) and Flocco et al. (2015). It relies on the following set of equations:

$$V_{\mathrm{pnd}} = h_{\mathrm{pnd}}f_{\mathrm{pnd}} \tag{28}$$

$$h_{\mathrm{pnd}} = \min(a_{\mathrm{emp}}f_{\mathrm{pnd}}, 0.9h_i) \tag{29}$$

$$\rho_w\frac{dV_{\mathrm{pnd}}}{dt} = (1 - r)\left[\rho_i\frac{dh_{\mathrm{top}}}{dt} + \rho_s\frac{dh_s}{dt} + F_{\mathrm{rain}}\right] \tag{30}$$

where $V_{\mathrm{pnd}}$ is the melt pond volume, $h_{\mathrm{pnd}}$ the pond depth, $f_{\mathrm{pnd}}$ the pond fraction, $\rho_i$ and $\rho_s$ the ice and snow densities, $a_{\mathrm{emp}}$ is the slope from a linear fit between the pond fraction and the pond depth from SHEBA observations (Perovich et al., 2003), and $r$ is the fraction of surface meltwater that runs off the ice floes and is not collected by the ponds.

Like in Holland et al. (2012), the melt pond volume is a virtual reservoir that does not impact the freshwater and salt fluxes with the ocean. Like Flocco et al. (2010) and Holland et al. (2012), we first estimate the change in $V_{\mathrm{pnd}}$ from the accumulated water using (30), then estimate the fraction of ponded ice $f_{\mathrm{pnd}}$ by combining (28) and (29). We then check that $h_{\mathrm{pnd}}$ is not



higher than $0.9h_i$ to obey (29); if this is the case, we reduce $h_{\mathrm{pnd}}$ by reducing $V_{\mathrm{pnd}}$ while keeping $f_{\mathrm{pnd}}$ untouched. This can

be interpreted as the vertical draining of ponds that are almost as deep as the ice is thick.

For the sake of simplicity, we use a constant albedo $\alpha_p$ for the melt ponds. To avoid having a significant effect from very

shallow ponds, we introduce a low limit threshold to the pond depth set to 0.05m and reduce $f_{\mathrm{pnd}}$ if needed.

In contrast to Holland et al. (2012) and Flocco et al. (2010), we consider $r$, the fraction of meltwater that runs off the ice,

to be constant instead of a function of sea ice concentration. This change is motivated by new insights from MOSAIC data

(Webster et al., 2022; Smith et al., 2024). Smith et al. (2024) suggest that, after an initial peak at $\simeq 40\%$, the fraction of surface

freshwater that is stored in ponds (i.e., $1 - r$) reduces to $\simeq 10\%$. They acknowledge that their data may be biased low due to

high lateral draining at the location of their measurements, close to the edge of a floe. However, they find relatively similar

results applying the same analysis to data from SHEBA, even though the fraction bounces back slowly to 20% by the end of the

summer. From these observations, a reasonable range for $r$ should be between $\simeq 60\%$ and $\simeq 90\%$, while it is $\simeq 20\%$ in compact

ice using the formulation in Holland et al. (2012) and Flocco et al. (2010).

Since neXtSIM lacks an explicit representation of lateral and vertical draining, obtaining albedo and melt pond fraction

values consistent with observations requires using a large value of $r$ to avoid accumulating too much water in ponds by the end

of the summer. Therefore, we recommend using a relatively high value for $r$ (e.g., $r \simeq 0.9$).

We also allow changing the $a_{\mathrm{emp}}$ value. In Holland et al. (2012) and Flocco et al. (2010), $a_{\mathrm{emp}}$ is a constant set to 0.8 from

Perovich et al. (2003), but the MOSAIC data presented in Webster et al. (2022) (Fig. 9 and 10) would suggest a higher slope

value ($a_{\mathrm{emp}} \simeq 1$) than the one obtained doing linear fit on SHEBA data only. Low ponded-ice fractions associated with deep

ponds increase the value of $a_{\mathrm{emp}}$ and are also associated with draining, hence strongly depending on the location. Again, as

draining is missing in our model, it is possible to increase the value of $a_{\mathrm{emp}}$ so that deeper ponds are not associated with large

ponded-ice fractions.

In freezing conditions, a lid forms over the pond, which reduces the amount of meltwater in the pond:

$$\frac{dV_{\mathrm{lid}}}{dt} = \frac{Q_{\mathrm{ia}}}{\rho_i L_f} \tag{31}$$

$$\frac{dV_{\mathrm{pnd}}}{dt} = -\frac{dV_{\mathrm{lid}}}{dt}\frac{\rho_i}{\rho_w}, \tag{32}$$

where $V_{\mathrm{lid}}$ is the change in lid sea ice volume, $Q_{\mathrm{ia}}$ is the heat lost by the ice to the atmosphere, and $L_f$ is the latent heat of

fusion. If the meltwater under the lid keeps freezing:

$$\frac{dV_{\mathrm{lid}}}{dt} = \frac{\min(Q_{\mathrm{ia}} - Q_{\mathrm{ic}}, 0) + Q_{\mathrm{ic}}}{\rho_i L_f f_{\mathrm{pnd}}}, \tag{33}$$

with $Q_{\mathrm{ic}}$ the heat lost by the meltwater in the pond:

$$Q_{\mathrm{ic}} = \frac{T_{\mathrm{pnd}} - T_i}{k_i V_{\mathrm{pnd}}/f_{\mathrm{pnd}}} \tag{34}$$

with $T_{\mathrm{pnd}}$ the temperature of meltwater in the pond that we assume to be equal to the freezing temperature of water with the

same salinity as sea ice (constant in neXtSIM). If the ice lid thickness $V_{\mathrm{lid}}/f_{\mathrm{pnd}}$ reaches 1 m, the melt pond and its lid are

removed ($V_{\mathrm{pnd}} = V_{\mathrm{lid}} = 0$).



In the end, the total albedo $a_{\text{total}}$ is equal to:

$$a_{\text{total}} = A_0 a_{\text{ow}} + a_{\text{cons\_ice}} + a_{\text{young\_ice}}, \tag{35}$$

where $a_{\text{ow}}$ is the albedo of open water, $a_{\text{cons\_ice}}$ the total albedo of the consolidated ice-covered area and $a_{\text{young\_ice}}$ the total albedo of the young ice-covered area. These terms are computed as follows:

$$a_{\text{cons\_ice}} = (A - f_s - f_{\text{pnd}})a_i + f_s a_s + f_{\text{pnd}} a_{\text{surf\_pnd}} \tag{36}$$

$$a_{\text{young\_ice}} = (A_y - f_{s,y})a_i + f_{s,y} a_s, \tag{37}$$

with $a_i$, $a_s$ the albedos of bare ice and snow, respectively, and $f_s$ and $f_{s,y}$ the fractions of consolidated and young sea ice covered by snow, which are estimated as:

$$f_s = \frac{h_s}{h_s + 0.02} \tag{38}$$

$$f_{s,y} = \frac{h_{s,y}}{h_{s,y} + 0.02}, \tag{39}$$

where $h_s$ and $h_{s,y}$ are the snow thickness on consolidated and young ice, respectively. In (36), $a_{\text{surf\_pnd}}$ is the albedo at the surface of the pond (including the lid), which we compute as:

$$a_{\text{surf\_pnd}} = a_i + (a_{\text{pnd}} - a_i)e^{-\kappa * V_{\text{lid}}/f_{\text{pnd}}}, \tag{40}$$

with $\kappa$ a constant related to the extinction coefficient of sea ice. Taking $\kappa = 4$ and $a_i = 0.76$ gives a dependency of the lid albedo to its thickness, similar to the logarithm law suggested by Ebert et al. (1995).

### 3.3 Ridge ratio calculations

NeXtSIM tracks the volume ratio of ridged ice throughout the simulation. Convergence in the Lagrangian framework is represented by a shape change of the model element, while mass in the element is conserved. When this happens, the mean thickness in the element increases, which can be ascribed to ridging. This is easily converted to a volume fraction, which can then be tracked. Ridges only form when the element is fully ice-covered; otherwise, the convergence reduces the open water fraction. However, the following equations also hold when the concentration is less than 100%.

Tracking the volume ratio in the Lagrangian reference frame requires no assumptions on ridge characteristics such as shape, keel depth, sail height, or frequency, while some assumptions about thermodynamic melt and growth of ridges must be made. Taking into account the ridging of young ice (when using the young-ice class), four cases must be considered and associated assumptions must be made:

1. Ridging of consolidated ice: Assume the conservation of mean thickness of level ice

2. Ridging of young ice: Assume the conservation of volume of level ice

3. Thermodynamic growth: Assume the conservation of the volume of ridged ice





4. Thermodynamic melt: Assume that ridged and level ice melt at the same rate

When ridges form in consolidated ice due to convergence, we assume that the mean thickness of level ice is conserved. This can be written as

$$(1 - R^n)\frac{H^n}{A^n} = (1 - R^{n+1})\frac{H^{n+1}}{A^{n+1}}. \tag{41}$$

Here, $R$ is the volume ridge ratio, $H$ is the mean ice thickness over the element (or volume per unit area), and $A \in [0, 1]$ is the fractional ice concentration in the grid cell. The superscripts $n$ and $n + 1$ denote the current and following time steps,

respectively. This equation can be rewritten as

$$R^{n+1} = 1 - (1 - R^n)\frac{A^{n+1}}{A^n F}, \tag{42}$$

where $F = H^{n+1}/H^n$. Ridges only form when $A^n = A^{n+1} = 1$, but the above equation holds in general due to the way $A$ changes under Lagrangian advection (see section 4.1). The ratio $F$ can also be related to the Lagrangian change in element area, $S$, since volume conservation

$$S^n H^n = S^{n+1} H^{n+1} \tag{43}$$

can be written as

$$\frac{S^n}{S^{n+1}} = \frac{H^{n+1}}{H^n} = F. \tag{44}$$

When using the young-ice class, we assume that ridged young ice is added to the consolidated ice class without affecting the mean thickness of level ice in that class. This can be written the same as equation (41), but in this case, we always have

$A^n = A^{n+1}$, and the change in $H$ is due to mechanical redistribution, i.e. $H^{n+1} = H^n + \Psi_y$, where $\Psi_y$ is the thickness of ice transferred from the young-ice class due to convergence (see appendix A2 of Rampal et al., 2019). The resulting evolution equation for $R$ is therefore

$$R^{n+1} = 1 - (1 - R^n)\frac{H^n}{H^n + \Psi_y}. \tag{45}$$

For thermodynamic growth and melt, we first assume that ridged ice does not grow thermodynamically; thus, all ice added

through thermodynamic growth is level. This is a reasonable assumption, although it does not consider the consolidation of ridges. This assumption can be interpreted as the conservation of the volume of ridged ice

$$R^n H^n = R^{n+1} H^{n+1}, \tag{46}$$

or

$$R^{n+1} = R^n H^n / H^{n+1}. \tag{47}$$

For melt, we assume that both ridged and level ice melts at the same rate. This assumption leads us to conserve the ridge ratio at melt, i.e. $R^{n+1} = R^n$. This assumption is not entirely accurate, as ridges are known to melt faster than level ice under certain



circumstances (e.g. Rigby and Hanson, 1976; Skyllingstad et al., 2003; Salganik et al., 2023). There is, however, considerable uncertainty related to how ridges change during the melt season, so we content ourselves with the simple assumption of conserving the ridge volume ratio during the melt.

## 3.4  Ice age tracers

Regan et al. (2023) introduced the tracers $A_{my}$ and $H_{my}$, which are, respectively, the concentration and volume of multi-year ice (MYI) - defined as ice that survived the summer melt (for a full description of these tracers, see Regan et al. (2023).) They are affected by the following processes:

- *Freezing* contributes to an increase in FYI volume and concentration only.

- *Melting* acts as a sink term for both FYI and MYI concentration and volume.

- *Replenishment* of MYI occurs when the ice in a mesh element has undergone three consecutive days of mean growth following the height of the melt season (set as 1 August).

- *Convergence*, through ridging, of ice acts as a sink term for area only, not affecting volume.

An average sea ice age estimate was not included in the study of Regan et al. (2023). We estimate the sea ice age by a surface

average age ($\alpha_A$) and a volume average age ($\alpha_V$). The first is intended to relate to the age deduced from satellite observations, while the second is a more physical measure of the total age of the ice. The volume-averaged age tracer follows the same ideas as, e.g. Hunke and Bitz (2009), while we are not aware of an implementation of an area-averaged ice age tracer in the community.

The two tracers evolve according to

$$\alpha_A^{n+1} = w_A(\alpha_A^n + \Delta t) + (1 - w_A)\Delta t, \tag{48a}$$

$$w_A = \min\left\{1, \frac{A^n}{A^{n+1}}\right\}, \tag{48b}$$

$$\alpha_V^{n+1} = w_V(\alpha_V^n + \Delta t) + (1 - w_V)\Delta t, \tag{48c}$$

$$w_V = \min\left\{1, \frac{H^n}{H^{n+1}}\right\}. \tag{48d}$$

The definitions of $w_V$ and $w_A$ ensure that in melting conditions ($H$ and $A$ are decreasing), the age increases by $\Delta t$, while in

freezing conditions, new ice is given the age $\Delta t$ and the weights $w_V$ and $w_A$ are the fractions of ice determined by volume or area (respectively ) that were already present at the previous time step.





## 4 Lagrangian mesh

### 4.1 Lagrangian advection and remeshing

NeXtSIM uses a triangular mesh with moving nodes and a remeshing scheme, as previously documented in Bouillon and
Rampal (2015); Rampal et al. (2016b); Samaké et al. (2017). In this section, we will give an overview of the Lagrangian
advection used in neXtSIM, as well as detailing the conservative remapping scheme added since Samaké et al. (2017).

The neXtSIM mesh is an unstructured triangular mesh on an $\{x, y\}$-plane, usually a polar-stereographic plane, although
other projections are technically possible. The initial model mesh can be constructed using tools such as Gmsh (Geuzaine
and Remacle, 2009) or any mesh generator that can produce a mesh in the Gmsh format. The finite element method used in
neXtSIM stipulates that velocities are calculated on the nodes of the elements while tracers are computed on the element. The
Lagrangian advection then consists of the following steps:

1. Move the nodes of the mesh by $\Delta \mathbf{x} = \Delta t \mathbf{u}$

2. Modify concentration and thickness values to conserve volume and area

3. Check the distortion of the resulting mesh

4. If the mesh is too distorted, then

    (a) Adapt the mesh where it has become too distorted

    (b) Copy model fields from old to new mesh, interpolating where new elements are introduced

    (c) Partition the mesh

Moving the mesh nodes by the displacement $\Delta \mathbf{x} = \Delta t \mathbf{u}$, where $\Delta t$ is the model time step and $\mathbf{u}$ is the ice velocity, is trivial.
Nodes on land boundaries are fixed with $\mathbf{u} = 0$, and while we calculate the velocity at the nodes at open boundaries, those
nodes are also kept fixed.

Once the nodes have moved, the area of the elements, $S$, changes. To conserve the ice and snow volume and the ice area
in each element, these prognostic variables must be multiplied with the fraction $F = S^n / S^{n+1}$, where $n$ and $n+1$ denote the
state before and after the nodes are moved. This follows directly from volume and area conservation

$$S^n H^n = S^{n+1} H^{n+1}, \tag{49}$$

$$S^n A^n = S^{n+1} A^{n+1}, \tag{50}$$

where $H$ denotes snow or ice volume and $A \in [1, 0]$ the concentration fraction. These are the only prognostic variables affected
by the change in element area due to the Lagrangian advection. In- and out-flux through open boundaries is ensured by keeping
these nodes fixed and not updating these prognostic variables on elements bordering an open boundary. This means that ice
will flow out as if there was no resistance and flow in as if the ice state outside the boundary was the same as that inside it.





The mesh adaptation is made using a port of the bi-dimensional anisotropic mesh generator, bamg (Hecht, 1998; Larour et al., 2012). The main advantage of bamg is that it can adapt the mesh only where it is too deformed, leaving other nodes in their original position. The mesher in bamg returns C++ objects with complete element and node connectivity, making it possible to quickly address nodes and elements connected to any given node or element of the mesh. The mesh adaptation is described in more detail in Rampal et al. (2016b).

In previous versions of neXtSIM (i.e. Rampal et al., 2016b), fields were mapped from the old to the adapted mesh using an algorithm which identified areas where the two differed ("cavities") and then interpolated the fields from the old mesh to the cavities of the new mesh in a conservative manner. This approach was very efficient and highly optimised, but ultimately very complex and occasionally fragile. When developing the coupling routines to couple neXtSIM with an ocean model (see section 4.3), we created a more robust conservative remapping approach, described in the following subsection. We could use this approach to map between the old and adapted meshes with only minimal code changes, and the relative simplicity of the scheme and its robustness, in addition to substantial code reuse, more than made up for the fact that the new scheme is less efficient than the old one.

While the model is parallelised using MPI (Samaké et al., 2017), the remeshing process is not parallelised. Therefore, all mesh information and model fields must be gathered to one MPI processor to adapt the mesh, remapping from the old to the new mesh and partitioning the new mesh, which is then scattered to the other MPI processes afterwards. The mesh is partitioned using the METIS partitioner.

Doing the mesh adaptation and field remapping on a single MPI process generates a substantial bottleneck for large problems, even if the Lagrangian advection itself is highly efficient. This was already identified by Samaké et al. (2017), but their solution of using an arbitrary Lagrangian-Eulerian (ALE) approach was technically challenging and was abandoned. As a result, neXtSIM is, in practice, limited to meshes with $\mathcal{O}(10^5)$ elements running on $\mathcal{O}(100)$ MPI processors. For larger problems, the model mostly stops scaling with the number of MPI processors, meaning that the execution time stays the same, regardless of how many processors are used.

## 4.2 Conservative remapping

NeXtSIM uses a conservative remapping scheme, where values from a donor mesh are mapped to a receiver mesh using weighted averaging with the area of overlap between elements in the two grids as weights. The weights are calculated only once and then applied to many fields at a very low computational cost. The weight calculation algorithm below relies on two features of bamg: the complete connectivity information of the mesh and a bamg utility, which allows one to find the element covering any point in the domain. The algorithm is simple (see Algorithm 1), relying on only three checks for element nodes and element-element intersection and recursion. The algorithm returns two lists: a list of elements of the donor mesh overlapping a given element of the receiver mesh, $\mathcal{L}$, and a list of the area of overlaps, used as weights, $\mathcal{W}$. In practice, a single check routine is called for the receiver element to build $\mathcal{L}$ and $\mathcal{W}$.

The first step of the algorithm is to find an element in the receiver mesh, $\mathcal{E}_r \in \Omega_r$, that overlaps with the barycentre of a given element in the donor mesh, $B(\mathcal{E}_d) \in \Omega_d$. This is done using an interpolation utility in bamg, which uses the underlying





---

**Algorithm 1** The algorithm behind the recursive function to generate a list, $\mathcal{L}$, of donor elements, $\mathcal{E}_d \in \Omega_d$, overlapping the receiver element $\mathcal{E}_r$, and the area of overlap, $\mathcal{W}$. $\mathcal{P}$ is a temporary list of points. $N$ is a function returning the nodes of an element, $I$ is a function returning the element number, $X$ is a function returning the intersection point and vertices, $\mathcal{V}$, of two elements, and $A$ is a function returning the area of the polygon formed by the list of points given on input.

---

**if** Any of $N(\mathcal{E}_d) \in \mathcal{E}_r$ **then**
    Add all $N(\mathcal{E}_d) \in \mathcal{E}_r$ to $\mathcal{P}$
    **for all** $\mathcal{E}'_d \in \Omega_d$ sharing $N(\mathcal{E}_d)$ **and** $I(\mathcal{E}'_d) \notin \mathcal{L}$ **do**
        Call self for $\mathcal{E}'_d$
        Append to $\mathcal{L}$ and $\mathcal{W}$ from recursive call
    **end for**
**end if**
**if** Any of $N(\mathcal{E}_r) \in \mathcal{E}_d$ **then**
    Add all $N(\mathcal{E}_r) \in \mathcal{E}_d$ to $\mathcal{P}$
**end if**
**if** Any $X(\mathcal{E}_d, \mathcal{E}_r)$ exists **then**
    Add $X(\mathcal{E}_d, \mathcal{E}_r)$ to $\mathcal{P}$
    **for all** $\mathcal{E}'_d \in \Omega_d$ sharing $\mathcal{V}$ with $\mathcal{E}_d$ **and** $I(\mathcal{E}'_d) \notin \mathcal{L}$ **do**
        Call self for $\mathcal{E}'_d$
        Append to $\mathcal{L}$ and $\mathcal{W}$ from recursive call
    **end for**
**end if**
Add $I(\mathcal{E}_d)$ to $\mathcal{L}$
Add $A(\mathcal{P})$ to $\mathcal{W}$
**return** $\mathcal{L}, \mathcal{W}$

---

quadtree to find the element in a given mesh that overlaps a set of points in the domain. This is by far the most time-consuming part of the algorithm. Once all overlapping elements have been found, the algorithm loops through all donor elements, $\mathcal{E}_d$, performs three simple checks, and initiates checks on neighbour elements, recursively, as needed.

    The first check is to test if any of the nodes of the donor element are inside the receiver element, $N(\mathcal{E}_d) \in \mathcal{E}_r$. The list of element nodes, $N(\mathcal{E})$, is available as a bamg data structure, and the test is performed using a point inclusion in polygon test

(Franklin, 2023). This test is both accurate and extremely fast. It can also be used for any polygon shape, which is advantageous when using this algorithm to interpolate between a triangular mesh and a quadrangular grid (see section 4.3). Suppose a node of the donor element is inside the receiver element. In that case, this is registered in a list, $\mathcal{P}$, the element number, $I(\mathcal{E}_d)$ is added to the list of elements, $\mathcal{L}$, and the function performing the checks is called for each of the elements of the donor mesh which shares the node.





The second check tests if any of the receiver nodes are inside the donor element, $N(\mathcal{E}_r) \in \mathcal{E}_d$. The procedure is identical to the one in the first test, except that no recursive call is necessary in this case. Any $I(\mathcal{E}_r)$ are added to $\mathcal{L}$ and $N(\mathcal{E}_r)$ to $\mathcal{P}$.

The third check is to test if any of the vertices of $\mathcal{E}_d$ and $\mathcal{E}_r$ intersect. If they do, the intersection point $X(\mathcal{E}_d, \mathcal{E}_r)$ is added to $\mathcal{P}$. The intersection detection is quick and can be done for any two line segments, so the algorithm can be used for both triangular and quadrangular meshes. If the vertices of $\mathcal{E}_d$ and $\mathcal{E}_r$ intersect, then the function calls itself to perform the checks

on the element in $\Omega_d$ sharing the intersecting vertex, $\mathcal{V}$, with $\mathcal{E}_d$.

After these three checks, $\mathcal{P}$ is a list of points delineating a polygon of the overlap between $\mathcal{E}_d$ and $\mathcal{E}_r$. The algorithm ends by adding the element number, $I(\mathcal{E}_r)$, to the list $\mathcal{L}$ and the area of the polygon, $A(\mathcal{P})$, to the list $\mathcal{W}$. The area calculation is done using the shoelace formula (also known as Gauss's area formula or the surveyor's formula, e.g. Braden, 1986). Note that because of the recursive calls within the first and last checks, the lists $\mathcal{L}$ and $\mathcal{W}$ will be fully populated by all overlapping

elements at this point. Additional checks and early exits may be implemented when all $N(\mathcal{E}_d) \in \mathcal{E}_r$ or all $N(\mathcal{E}_r) \in \mathcal{E}_d$. This gives a marginal improvement in execution time. Once the lists $\mathcal{L}$ and $\mathcal{W}$ are obtained, calculating the field values in $\mathcal{E}_r$ from the elements $\mathcal{E}_d$ listed in $\mathcal{L}$, using the weights from $\mathcal{W}$, is trivial.

### 4.3   OASIS coupling

NeXtSIM can be coupled to both ocean and wave models through the OASIS coupler (e.g. Boutin et al., 2021, 2023). OASIS

(Craig et al., 2017) is a coupler that requires minimal code modification in the models being coupled and is very well suited to coupling substantially different code bases, as is the case here. To couple different models, each model must implement initialisation and finalisation routines and include the OASIS `get` and `put` function calls for receiving and sending coupling fields.

The OASIS interface implementation in neXtSIM is unusual because of the moving mesh neXtSIM uses. OASIS is designed

for fixed meshes, and the OASIS initialisation routine should specify the mesh shape, coordinates, and domain decomposition. This allows OASIS to calculate interpolation weights and the communication pathways between the MPI domains of the different models once at startup. In the case of neXtSIM, the mesh and domain decomposition changes continuously throughout the simulation. Therefore, the OASIS interface in neXtSIM is set up so that the two models communicate through a fixed exchange grid within neXtSIM and on neXtSIM's root MPI processor.

For most setups, the exchange grid is identical to the grid of the other (ocean or wave) model. When coupling to an ocean model, the ice and ocean models must share an identical coastline to ensure the conservation of fluxes between the two models. The initial mesh of neXtSIM is then constructed to exactly trace the coastal boundaries of the exchange and ocean grid. NeXtSIM then uses the conservative remapping algorithm (section 4.2) to interpolate between the moving mesh and the fixed exchange grid. To avoid recalculating the exchange weights every time step (as per section 4.2), the mesh is considered sta-

tionary in between remeshing steps.

The coupling procedure described above relies on several compromises in terms of both efficiency and accuracy. Recalculating the interpolation weights every time a remeshing occurs is obviously less efficient than only calculating them at the start of the run. The remapping algorithm we use is, however, efficient enough that this only takes about 5% of the run time in a





typical ice–ocean coupled setup (such as the one used by Boutin et al., 2023). Another efficiency loss is due to the fact that all
communication must go through the neXtSIM root MPI processor. It is difficult to assess this cost, but implementing proper parallel coupling was a significant step forward in the OASIS development (Craig et al., 2017). In a typical ice–ocean setup, about 12% of the total run time is spent in the OASIS `put` and `get` routines, but this includes time neXtSIM spends waiting for the ocean model. Some loss of accuracy is incurred because the mesh is considered to be stationary between remeshing steps. In setups at between 10 and 20 km resolution (e.g. Ólason et al., 2022; Boutin et al., 2023), remeshing occurs at between
three and six hourly intervals, resulting in displacement errors no larger than approximately 1/3rd of the model resolution.

## 4.4 Input/Output operations

NeXtSIM supports two types of outputs: snapshots of the model state with complete mesh information and netCDF output on a fixed, rectangular output grid. The first is used to output the exact model state and is also used for restart files. The second is used to generate temporal averages and to simplify post-processing and analysis, as it is the standard format used in
oceanography and climate science. The mesh-based snapshots are binary files of mesh and model field values, accompanied by ASCII files with the file header information. We have developed a library of Python routines called pynextsim[1] to read, analyse, and modify the mesh-based output from neXtSIM.

In its simplest incarnation, the neXtSIM netCDF output uses a nearest-neighbour interpolation from the neXtSIM mesh to the output grid specified for the netCDF file. At each time step, the output field values in the element covering the centre point
of each output grid cell are added to that grid cell's value, and the average is then written to file at the given output interval. This sampling technique provides the same kind of results as moorings in the real ocean. The prefix of the netCDF outputs and functions in the code draw their names from this similarity. Finding the overlap between the model mesh and the output grid is time-consuming, so we assume the mesh is stationary between remeshing steps. This is the same approach as in the coupling code, which is based on the same underlying functions as the netCDF output code.

The common code base of the coupling functions and netCDF output functions allows us to take advantage of the conservative remapping capabilities developed for coupling and remeshing. When in coupled mode, the netCDF output grid is the same as the coupler exchange grid and instead of using a nearest-neighbour interpolation, the conservative remapping approach is used. This has the added benefit that the netCDF outputs are conserving, which is crucial for budget calculations (e.g. Ólason et al., 2021; Boutin et al., 2023). This capacity can also be used in stand-alone mode, provided a specially crafted initial mesh
and output grid definitions are provided, following the same requirements as for the mesh and exchange grid in the coupled case.

## 5 Simulation examples and use-cases

This section gives an overview of the model results and capabilities.

---

[1]https://github.com/nansencenter/nextsim-tools



## 5.1 An idealised test case

While neXtSIM was developed primarily to run large-scale simulations, the model can also be set up for idealised simulations
to explore the details of certain physical processes and their parameterisations. To illustrate this, we show the results of an
idealised experiment, exploring the difference between two damage update schemes. The two experiments consist of simulating
a 1000×1000 km ice block fixed on one side to a coast with ice thickness 1 m, concentration 100 %, damage 0, and wind 5 m/s
blowing towards the coast (see Figure 2.a). Both experiments run 24 hours on a mesh with ≈10 km resolution, at 5 s time step,

with 3 sub-steps. Other parameters are taken from Ólason et al. (2022) and Table 1. In one experiment, the stress and damage
are updated using the formulation from Ólason et al. (2022) (i.e., with $\Delta t/t_d$), and in the second one - using equations 8 and
10 (without $\Delta t/t_d$).

Figure 2.a shows the initial concentration and total ice motion over 24 hours, while panels b and c show total deformation
from the two experiments. Both deformation fields look somewhat similar; however, the results of the second experiment show

narrower deformation zones and more small-scale faults in sea ice. To illustrate the differences in the process of ice faulting,
we can analyse the propagation of the internal stress and damage in these experiments (see Figure 3 and the video in the
supplement material).

After 2 hours, the fields of internal normal stress ($\sigma_N$, Figure 3, a and f) show quite similar patterns of two elastic wave trains
propagating from the upper corners, where ice is attached to the coast and where the faulting started. Ice damage increases in

the elements where the normal and shear stress exits the Mohr-Coulomb envelope, and lines of enhanced ice damage appear
on the maps (Figure 3, b and e). On these maps, the main difference of the second experiment is in the narrower front of stress
waves and, consequently, more localised damage. Analysis of the normal and shear stress evolution (see Figure 3, c and f) can
help understand the reason for this difference.

Two points (marked by blue and red colour in Figure 3, b and e) are located well outside the region of faulting and stress

behaviour (shown by blue and red trajectories in Figure 3, c and f) is quite similar. Two other points (marked by orange and
green dots) are placed in the faulting zone. The orange dot experienced ice damage (and stress relaxation) only once, while
the green one underwent several cycles of damage, stress relaxation, and stress build-up. Here, the stress trajectories differ
significantly.

In the first experiment, the $\Delta t/t_d$ factor is always below one, and once the stress tensor of the element exits the fault

envelope, the damage begins to increase slowly, and the stress begins to decrease slowly. Thus, the stress tensor returns to the
envelope in more than one sub-step, manifested in a curved trajectory of normal/shear stress going outside the envelope for
several steps and sub-steps. At the same time, as stress is still relatively high and damage is not high enough, the stress starts
to build up in the neighbouring elements, creating a broad front of the elastic wave. The broad front, in turn, leads to broader
lines of increased damage.

In the second experiment $\Delta t/t_d = 1$, the damage increases sufficiently quickly to allow the stress to return to the envelope
in one sub-step. That pushes the stress tensor trajectory on the envelope, creating narrower elastic waves with better damage




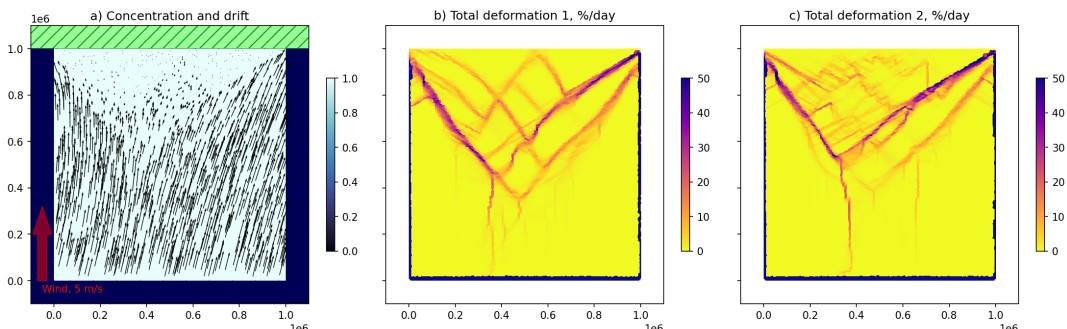

**Figure 2.** (a) Experiment setup for an idealised experiment. The domain is $100 \times 100$ km with a constant wind of 5 m/s blowing along the positive y-direction.

localisation. Such rapid change, however, tends to generate instabilities, leading to noisier stress and damage fields and the appearance of small-scale deformation lines.

These results demonstrate how using a dynamic timestep shorter than the damage propagation time, i.e. $\Delta t < t_d$, allows stresses in the model to temporarily exceed the failure envelope. This results in less localised damage and deformation fields. These supercritical states are short-lived but affect the simulation results, nonetheless. Supercritical states are not physical but could be considered acceptable if the formulation improved model stability without significantly impacting the results. The numerical stability of the model in large-scale setups does, however, not appear to be affected by the formulation of equations (8) and (9), even though the idealised tests do show increased instabilities when omitting $\Delta t/t_d$. The large-scale deformation in our 20 km resolution runs is also not affected to any substantial degree. In neXtSIM, we choose the formulation in equations (8) and (9) to avoid the unphysical supercritical states.

## 5.2 Stand-alone 15 year simulation

As a large-scale sea ice model, neXtSIM's main goal is to represent the evolution of sea ice's large-scale properties most accurately, for time scales from a few days to decades. Here, we run a 15-year-long simulation of the model in a stand-alone configuration to illustrate the model's capacity to represent different aspects of sea ice properties in the context of a stand-alone simulation.

As in Ólason et al. (2022), we use the hourly ERA5 reanalysis (Hersbach et al., 2020) for atmospheric forcing and the TOPAZ4 reanalysis (Sakov et al., 2012) for oceanic forcing. Initial sea ice thickness and concentration are set from the PIOMAS reanalysis (Zhang and Rothrock, 2003). Initial sea ice damage is set to zero. We use the same domain as Ólason et al. (2022), but with a coarser horizontal resolution of 20 km (instead of 10 km). Key model parameters that differ from Ólason et al. (2022) or were introduced in this study are summarised in 1.

To evaluate the consistency of neXtSIM's results, we compare them with various datasets of observed quantities. For sea ice extent, we use sea ice concentration from the climate data record of the EUMETSAT Ocean and Sea Ice Satellite Application



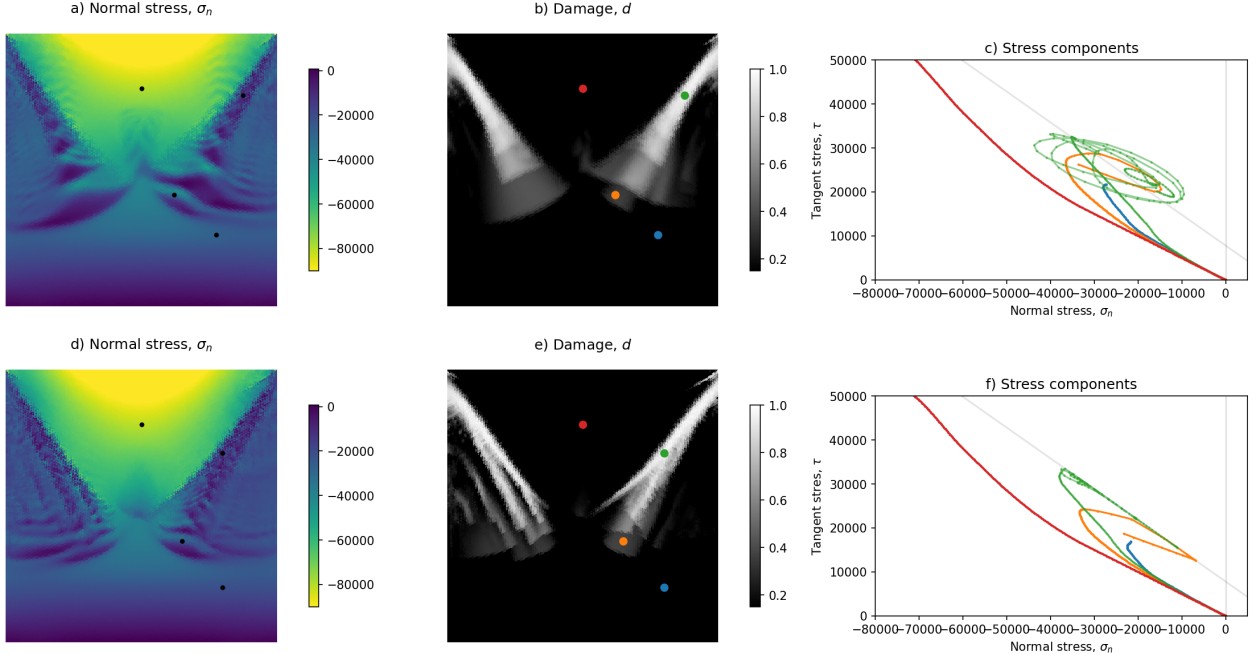

**Figure 3.** Normal stress, damage, and stress evolution at selected points of the domain using the two different damage and stress update schemes discussed in the text; with $\Delta t/t_d$ (a, c, and e) and without $\Delta t/t_d$ (b, d, and f).

**Table 1.** Parameters used for the 15-year-long simulation in this study that differ from Ólason et al. (2022) (or earlier publications) or that are introduced in this study for the first time.

| Parameter | symbol | new value | value in previous neXtSIM studies |
|---|---|---|---|
| Exponent of the thickness dependency for the ridging threshold | $n$ | 2 | 1.5 |
| Cohesion at the reference scale | $c_{\text{ref}}$ | 1.35 MPa | 2 MPa |
| Sea ice (bare) albedo | $a_i$ | 0.76 | 0.64 |
| Snow albedo | $a_s$ | 0.9 | 0.85 |
| Ratio of ice-ocean heat flux used to melt ice laterally | $\Phi_m$ | 0.2 | 0.5 |
| Melt pond albedo | $a_{\text{pnd}}$ | 0.3 | new |
| Fraction of surface meltwater that runs off the ice | $r$ | 0.92 | new |
| Extinction coefficient of sea ice | $\kappa$ | 4 m$^{-1}$ | new |
| Slope of the linear fit between pond fraction and depth | $a_{\text{emp}}$ | 0.8 | new |



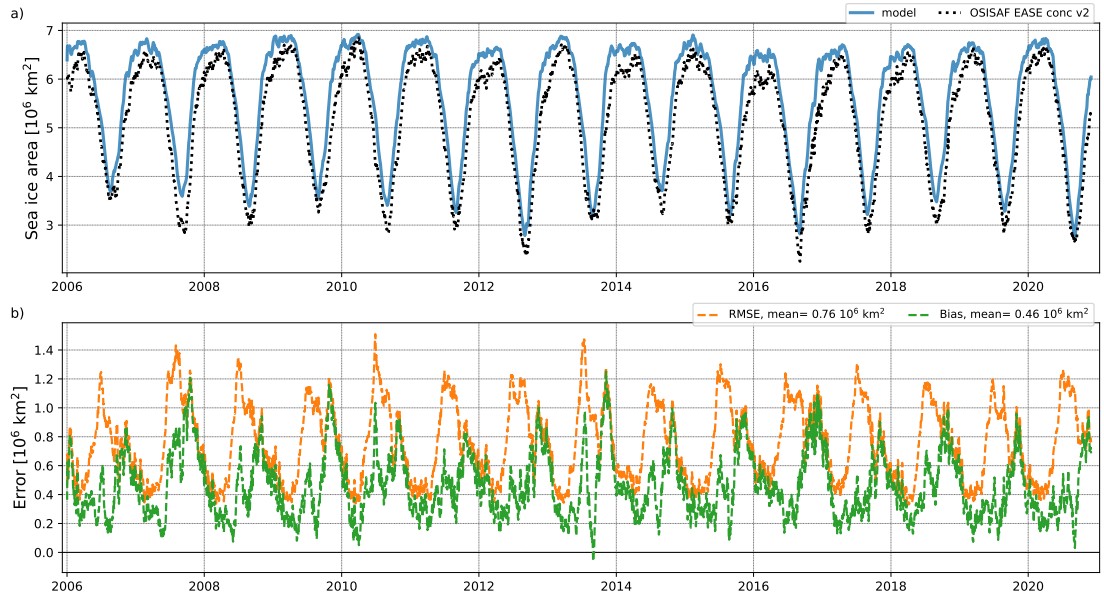

**Figure 4.** (a) Modelled and observed (OSI-SAF CDR) sea ice extent evolution for the period from January 2006 to December 2021. (b) IIEE and bias between the modelled sea ice drift and the OSI-SAF CDR dataset over the same period. The domain considered is shown in Figure 5e.

Facility (OSI-SAF, Lavergne et al., 2019). For sea ice volume and thickness, we use the dataset combining the observations

retrieved from the CryoSAT-2 and SMOS satellites, referred to as CS2SMOS (version 2.6, Ricker et al., 2017). For sea ice drift, we use the OSI-SAF climate data record (v1.0, Lavergne and Down, 2023). We use the dataset from the University of Bremen (Pohl et al., 2020; Istomina et al., 2023) retrieved using Sentinel-3 and ENVISAT for sea ice albedo and melt ponds. In our analysis, we generally compare model results and observations using the bias and RMSE. For sea ice extent, the bias is computed as described in Williams et al. (2021), and like them, we use the Integrated Ice Edge Error (IIEE, Goessling et al.,

2016) instead of the RMSE as it provides more information about the model's capacity to capture the extent evolution. We use the same spatial domain for integrated quantities as Boutin et al. (2023), which includes most of the Arctic Ocean but excludes the Greenland Sea and seas south of Bering Strait. This is because we mainly focus on pack ice, for which neXtSIM was originally developed (Rampal et al., 2016a).





**Figure 5.** (a) Modelled and observed (CS2SMOS) sea ice extent evolution for the period from January 2006 to December 2021. (b) RMSE and bias between the modelled sea ice drift and the CS2SMOS dataset over the same period. (c,d) show the 2010–2021 November to April sea ice thickness climatology in the model and as estimated by CS2SMOS. (e) is the climatology of the sea ice thickness bias between the model and CS2SMOS. These climatologies are computed from monthly averaged files. The black dashed contour in (e) shows the domain used to compute integrated quantities in section 5.2 (e.g., in panels a,b here).



### 5.2.1 Extent and volume

Sea ice extent and volume are generally the first quantities to be evaluated due to their effect on climate. As shown in Ólason et al. (2022) and Boutin et al. (2023), neXtSIM can generally simulate their evolution in a way consistent with observations (Figure 4 and 5).

As in the ice-ocean coupled setup presented in Boutin et al. (2023), sea ice extent is generally consistently captured (Figure 4). IIEE peaks at the end of summer, when sea ice extent is minimal, but remains generally under 1M km$^2$. It is associated
with a positive bias, meaning the simulation overestimates the yearly minimum extent. IIEE is lower in winter, as most of the domain used for its computation is covered by stopped ice. As in Boutin et al. (2023), including the totality of the domain has little qualitative effect on the IIEE evolution.

Modelled sea ice thickness also shows reasonable agreement with data from CS2SMOS (Figure 5). The interannual variability is visibly captured, and the slope corresponding to sea ice growth during the freezing season generally agrees with
observations. The bias and RMSE both grow from autumn to the end of winter (Figure 5b). In all years but 2011, the simulation first underestimated the amount of ice left at the end of the summer but then ended up growing more ice than in observations by April, when observations stopped being available. Looking at the thickness distribution, the general spatial patterns are captured, with thicker ice north of Greenland where the oldest ice is expected to be found (Figure 5c,d), and thinner ice in areas generally covered by first-year ice. The difference in sea ice thickness distribution reflects the behaviour of bias
and RMSE (Figure 5e), with the thickness of older and thicker ice underestimated while the thickness of younger and thinner ice is overestimated. This is a typical bias of sea ice models (Watts et al., 2021), and results can be improved by tuning some parameters, like the albedo, that are not very constrained and have a large impact on the amount of ice surviving the summer. Thinner ice thickness is more sensitive to parameters like the maximum thickness of the young ice class.

### 5.2.2 Drift

Sea ice drift is also an important metric to assess the quality of a sea ice model, as it significantly impacts sea ice balance through its transport of sea ice and its export through the Fram Strait (in the Arctic). A good representation of sea ice drift also improves the sea ice thickness distribution by properly capturing the location of older and thicker ice (Regan et al., 2023).

The simulation shows an agreement comparable to previous studies using neXtSIM (Rampal et al., 2016b; Williams et al., 2021; Boutin et al., 2023), with a generally low RMSE (less than 4 km/day most of the year). Both short-term (a few days) and
long-term (seasonal and interannual) variabilities are captured. The bias with the OSI-SAF CDR dataset shows a seasonal cycle as it increases in the summer. However, satellite-derived observations are not available in the summer. Instead, the OSI-SAF CDR uses estimates from a free-drift model with high uncertainties. Lavergne and Down (2023) suggest these estimates may be biased low, meaning it is uncertain whether our simulation is biased in the summer or not.

The simulations used here do not use the stability-dependent drag coefficients introduced in section 3.1. This is because
taking atmospheric stability into account slightly increases the bias and RMSE between the modelled sea-ice drift and the OSI-SAF CDR dataset. We have been unable to determine why improving the physical representation in the model in this way





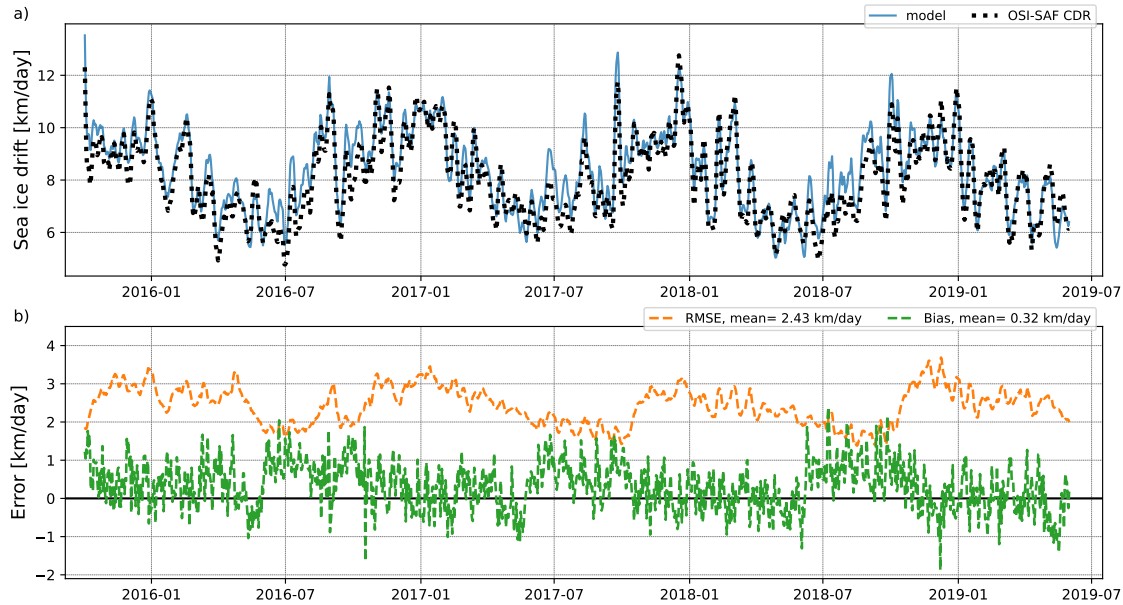

**Figure 6.** (a) Modelled and observed (OSI-SAF CDR) sea ice drift evolution for the period from October 2015 to June 2019. (b) RMSE (of drift speed) and bias between the modelled sea ice drift and the OSI-SAF CDR dataset over the same period. The domain considered is shown in Figure 5e.

gives worse results than the simpler approach of only using neutral drag coefficients. This issue will be investigated further in the near future.

### 5.2.3 Ridge ratio compared to observed sea-ice roughness

We compared the neXtSIM ridged ice thickness (computed as a product of ice thickness and ridged ice ratio) with ridging intensity from IceSat-2 (calculated as the ridge frequency per 1 km segment multiplied by the mean ridge sail height within the segment, Farrell et al., 2020). The maps in Figure 7 show remarkable similarity in depicting high roughness in perennial ice and coastal areas with first-year ice, with very low ice roughness in the Central Arctic. The correlation between the neXtSIM ridged ice volume and the IceSat-2 ridging intensity is also high (0.617, Figure 7), while correlations of other neXtSIM and

IceSAT2 products are lower (see Table 2).

### 5.2.4 Melt-pond fraction compared to satellite observations

This paper introduced the newly implemented melt pond scheme in neXtSIM and its effect on albedo in section 3.2. Here, we illustrate the behaviour of this scheme by comparing the seasonal distribution of the simulated sea ice albedo with estimates





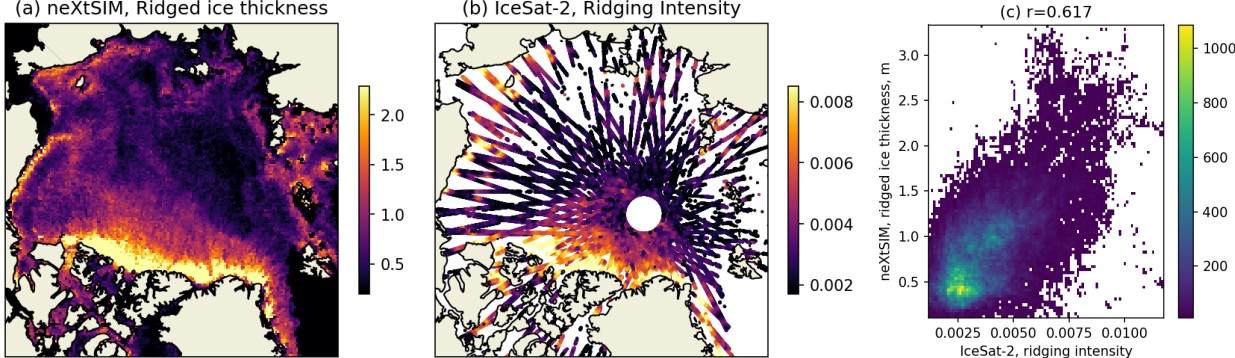

**Figure 7.** (a) Ridged ice thickness from neXtSIM on 2019-04-04, 12:00:00. (b) Ridging intensity from IceSat-2 orbits acquired between 2019-04-01 and 2019-04-07. (c) Scatter plot of IceSat-2 and neXtSIM products.

**Table 2.** Pearson's correlation between neXtSIM and IceSat-2 roughness-related products.

|  | Thickness | Ridged ice ratio | Ridge ice thickness |
|---|---|---|---|
| IS2 Roughness | 0.512 | 0.465 | 0.587 |
| IS2 Ridging intensity | 0.521 | 0.562 | **0.617** |

from the University of Bremen (Figure 8). We notice that both the magnitude and the distribution of albedo are generally

captured. This distribution is characterised by lower albedo at the margins of the sea ice cover and higher in the pack, with a strong latitude dependency in the observation dataset (see June in particular). The simulated albedo follows a general tendency but with a lower contract between pack ice and the ice edge and a generally lower albedo. This is partly explained when we look at the melt pond fraction (MPF) distribution (Figure 8). This distribution differs between observations and the simulation. Observed melt pond fraction shows large values close to the sea ice edge and lower values in the interior, while July and

August simulated MPF distributions are relatively uniform. We also notice that the modelled MPF is biased low in May–June but biased high later in August and September.

This behaviour of the MPF in the simulation can be explained mainly by simplifications we made in the parameterisation. The main simplifications are the absence of draining and the choice of a constant ratio of melted freshwater ending up in the ponds. Smith et al. (2024) suggest that this ratio is high early in the melt season (up to $\simeq 40\%$) and lower later (down to

$\simeq 10\%$). In the absence of draining, using a higher ratio would result in largely overestimated MPF by the end of summer, and the choice of a constant ratio has the advantage of simplicity. Adding a dependency on concentration, as in, e.g., Holland et al. (2012), who assume lower concentrations have a lower ratio due to higher draining, would only increase the current biases by increasing MPF in the pack and reducing it in the margins. In reality, sea ice topography, roughness, and porosity must play a



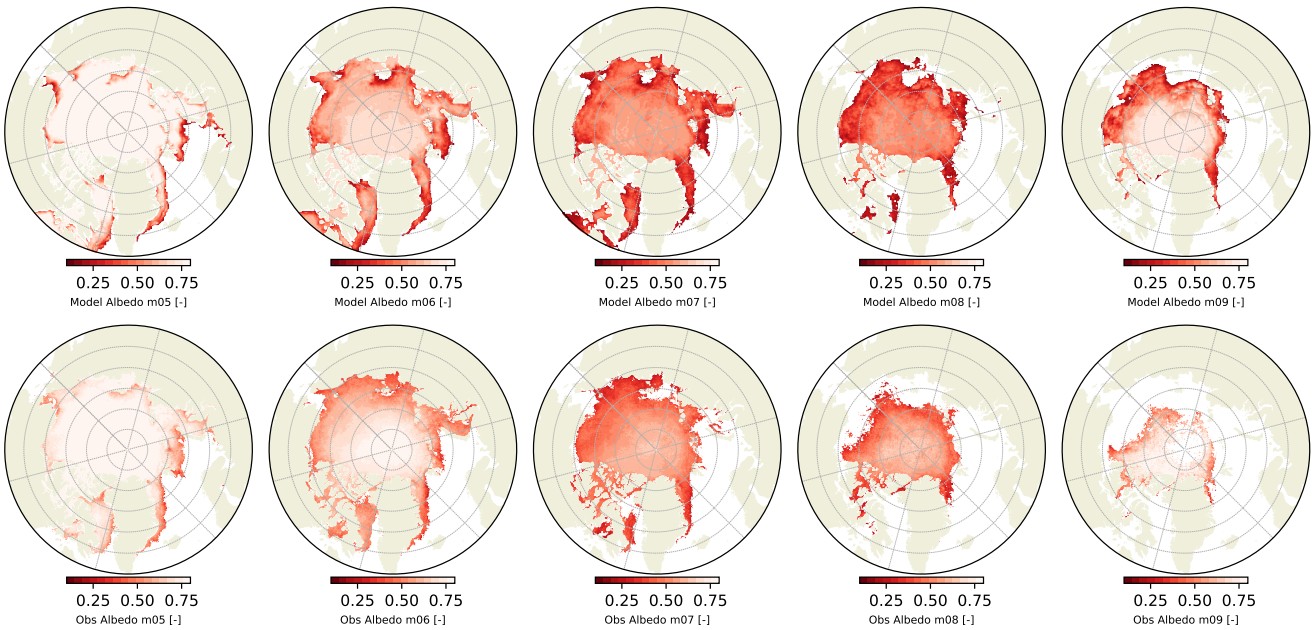

**Figure 8.** 2017–2021 climatology of the simulated and the observed broadband albedo for the months from May to September, when the observation product from Uni. Bremen is available.

significant role in both this ratio and the amount of draining, but these dependencies are not yet fully understood. Nevertheless,

activating the melt pond scheme albedo generally increases the sensitivity of simulated sea ice to melt (not shown), which helps reduce the extent of bias at the end of the summer. Still, it reduces the thickness of older ice, increasing the thickness bias we mentioned earlier. This effect is probably exacerbated by refrozen meltwater not counting towards ice mass in the model.

## 6 Summary and conclusions

This paper presents the latest version of the next-generation sea-ice model, neXtSIM. It also marks the first open-source release

of neXtSIM. This version depends on the core functionality already laid out in Rampal et al. (2016b), Samaké et al. (2017), and Ólason et al. (2022), but contains several significant new developments. The OASIS coupling interface and the associated conservative remapping algorithm are the most important technical developments. This was already used by Boutin et al. (2021, 2022, 2023); Regan et al. (2023), but the technical details are described here for the first time. The ridge ratio and ice age tracers are also currently part of the Copernicus Marine Service's Arctic sea-ice forecast[2], but only described here in detail

for the first time. The simulation results presented here also include more recent developments and physics implementations,

---

[2]https://doi.org/10.48670/moi-00004





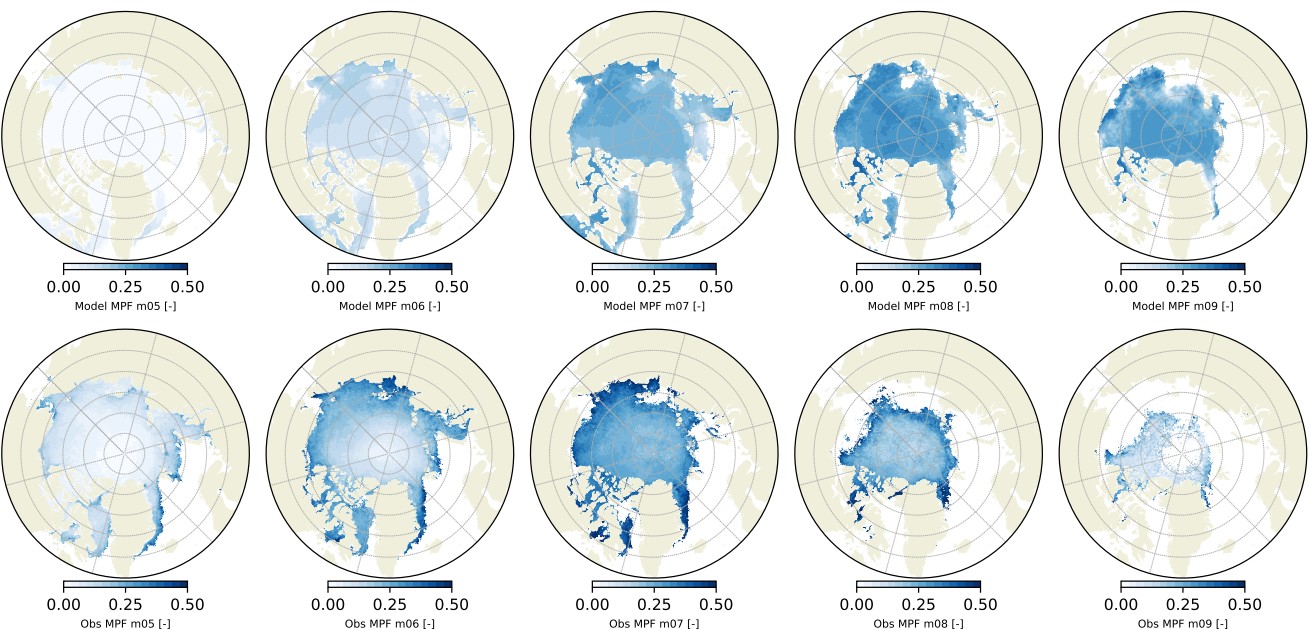

**Figure 9.** 2017–2021 climatology of the simulated and the observed melt pond fractions for the months from May to September, when the observation product from the University of Bremen is available.

such as the melt pond scheme and atmospheric drag scheme, which are yet to be used in scientific publications or forecast products.

*Code and data availability.* The neXtSIM code used for this paper (version 2.5.0) is available on Zenodo https://doi.org/10.5281/zenodo.14724536. The model is also available on https://github.com/nansencenter/nextsim. The code is released under the MIT licence.

The sea ice concentration dataset from OSI-SAF includes the OSI-450 sea ice concentration product available at ftp://OSI-SAF.met.no/reprocessed/ice/conc/ (last visited September 2024) and the interim version for data from 2021 onwards, OSI-430-b, available at ftp://OSI-SAF.met.no/reprocessed/ice/conc-cont-reproc/ (last visited September 2024). The sea ice drift climate data record is available at ftp://OSI-SAF.met.no/reprocessed/ice/drift_lr/v1/. The CS2SMOS sea ice thickness product is available at ftp://ftp.awi.de/sea_ice/product/cryosat2_smos/ (last visited September 2024). Albedo and melt pond data from the University of Bremen are available at https://data.seaice.uni-bremen.de/olci/ (last visited September 2024). ERA5 is available at https://www.ecmwf.int/en/forecasts/dataset/ecmwf-reanalysis-v5. TOPAZ4b reanalysis data are available at https://doi.org/10.48670/moi-00007. PIOMAS outputs are available at http://psc.apl.uw.edu/research/projects/arctic-sea-ice-volume-anomaly/data/model_grid (last visited September 2024).



*Author contributions.* EÓ led the writing and wrote most of the text. GB ran the 15 year simulation and wrote the sections on extent, volume, drift, and melt-pond fraction. TW developed and wrote up the ice age tracers. AK ran and wrote up the idealised test case and the ridge ratio
comparison section. DF, EÓ, and GB developed the melt pond parameterisation. RD and EÓ developed the drag parameterisation based on atmospheric stability. All authors contributed to the model development through direct code contributions or discussions of physics and parameterisations implementations or software development issues.

*Competing interests.* The authors declare that no competing interests are present

*Acknowledgements.* The development of neXtSIM has been supported by multiple projects, funded nationally in Norway, through European
collaborations, and internationally. The writing of this paper was supported by Norges Forskningsråd (grant no. 325292).





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
