# Peer review of "The next generation sea-ice model neXtSIM, version 2"

_EGUsphere, 2024_

## Author Comment (AC1)

**Response to comments from reviewer #1**

(Reviewer comments are in black, and our responses are in blue)

I welcome the effort to present the description of the current version of neXtSIM, but I think the paper needs to be improved in several directions before I can recommend it.

1. I miss a good description of the numerical part of the model, in particular its time stepping, details of the spatial discretization and the general workflow. The authors cite their previous work where I guess some explanation can be found. However, the value of technical papers in GMD is precisely that they provide an opportunity to document the model and describe details that would enable others to learn and to follow. Many quantities are left undefined in section 2. For example, the stress tensor appears in (1), and on line 54, but then disappears. Instead, there are undefined components, and finally there is the \sigma but without the boldface. Please be precise, it is almost impossible for a general reader to follow. The quantity K (bold face) is not defined, and \eta appears as sea surface height in (1), then as viscosity in (3), and \eta_0 is not specified. This continues in section 2.2, there are undefined strain rate invariants, the parameters e and P* are not specified, the equation for velocity is written (17), but no discrete equation for the stresses, and the reader is referred to other papers for \alpha, \beta and \beta' and \tau.

AR: We have added substantial details to the description of numerics and program flow. We added a new section, called "General model description", that provides a broad overview of the program flow. We also added subsections on spatial and temporal discretisation and included some details on the EVP and mEVP implementations. Indeed, the notation and general presentation in the dynamics section were not precise enough, and we have tried to improve this in the revised version. We appreciate the concrete examples from the reviewer, but we also found some other places where improvement was needed in this respect.

2. The presentation of section 3.1 should also be improved, and the iterative procedure for the drag coefficients needs to be presented in more detail.

AR: We don't use an iterative procedure for the drag coefficients in neXtSIM; it is not needed because we can advect the drag coefficients themselves with very little extra cost. We modified the text regarding this and hope it is clearer now. We also described the ice-class approach taken in neXtSIM in more detail, even though this has already been published elsewhere (in the spirit of the previous point). Barring any concrete examples of needed improvements, we found it difficult to improve the text further.

3. The explanation in section 4.2 needs a figure showing (defining) the geometry of the procedure.

AR: We have included an illustrative example with a figure that explains how the procedure works. It's not definitive; several other examples should be included for that. Even so, the example we chose touches on all aspects of the procedure and we believe that it should suffice so that the reader can understand other examples (e.g. with a triangle entirely inside a grid cell, a grid cell entirely covered by one triangle, a case where multiple triangle nodes are found within a grid cell, and the case of both meshes being made up of triangles).

4. Many technical steps in the model are related to the decision to use a moving mesh. First, it would be worthwhile to discuss why this choice is necessary and compare it with other possibilities. Remeshing should be equivalent to some diffusion, and if the choice made by the authors is motivated by the desire to keep the damage fields localized, the question is whether a comparable result can be achieved by using a high-order advection scheme. This would simplify many things (but make the advection more complicated). Second, I do not see any discussion of how mesh deformation will affect vector and tensorial quantities.

AR: We agree that a quantitative measure of the diffusion resulting from remeshing would be very useful. But we haven't come up with such a measure. In short, any stand test we are aware of involves translation or rotation, which the Lagrangian scheme will, by construction, handle fully conservatively. This leaves comparison with higher-order schemes, but this is contingent upon our implementing such schemes in neXtSIM. This is nontrivial and unlikely to yield a fair comparison, as we are not experts in advection schemes. Nonetheless, we have added a brief mention of the diffusion issue. Regarding how deformation affects vector and tensorial quantities, we modified the paragraph on deformation's impact on thickness and concentration to include this information. The end of that paragraph now reads:

> The ice concentration, thickness, and snow thickness are the only prognostic variables affected by changes in element area due to Lagrangian advection. This is because those are defined "per unit area", so when the area changes, their value must change. The value of all other advected variables (such as temperatures and stresses) remains unchanged. Velocities are defined on the nodes and are, therefore, not affected by the deformation of the element.

5. In the discussion of the test case (section 5.1) the authors compare the performance of two damage update schemes. While the field in 2c contains more detail, it is a discrete implementation and grid-scale details cannot be interpreted reliably. I would be suspicious of any features that are the size of several grid cells. Similarly, Fig. 3a and c show that there is grid-scale noise in normal stresses where damage tends to 1, meaning that the stresses are not really differentiable. Please, include discussion of these issues. From a numerical point of view, I would be concerned about the appearance of noise in solutions and consider some measures that restore smoothness.

AR: There is indeed grid-scale noise in the stress field, and to a lesser degree in the damage field as well. This is now briefly discussed in the revised text. While this noise is present, it does not impact the large-scale, observable quantities we like to evaluate the model against (such as drift, deformation, etc). Is it then better to add artificial terms to the equations to dampen the noise or to keep the physical formulation and tolerate some noise in the solution? This is, arguably, a question of modelling philosophy, and we have chosen not to try to dampen the noise, at least so far. I would also argue that we are not analysing grid-scale features in a noisy field, but rather how the grid-scale response has a global impact. It's perhaps a subtle distinction, but the point of this section is to briefly demonstrate how changing the damaging scheme affects the stress and damage states in the model. The impact is small, and it is even smaller in a realistic setup. But since this is a model description paper, we are more concerned with demonstrating what the model can do than delving deeply into the physics behind it. That should be left to a separate publication. We have appended a brief discussion of these points to the second-to-last paragraph of that section:

> … Such rapid change, however, tends to generate instabilities, leading to slightly noisier stress and damage fields. The way damage is modelled inevitably causes rapid, local

changes in the stress state, which can promote noise in the stress and damage fields, as is visible in figure 4. This noise, however, does not appear to affect large-scale observable metrics, such as deformation statistics, considered in e.g. Rampal et al. (2016b) and Ólason et al. (2022). We therefore have not attempted to damp this noise. It is, however, interesting that the form of damage evolution affects the level of noise in the simulation. Further exploration of the origin, impact, and potential damping methods for this noise is left to future study.

6. I also miss some general discussion of how the BBM and EVP compare with each other in both the simulated sea ice state and the requirements on time step and computational expenses. This would help the reader to see the differences and decide where the BBM rheology leads to advantages.

AR: We have added a short section at the end of the paper on the interest of using neXtSIM in general. This addresses all the reviewer's points, while putting them in the broader context of the neXtSIM model.

Minor points:

3,4 'but fails ...' But should it? At these resolutions the scales where truncation errors are small are perhaps 25 km or larger, and these scales are already too large for most leads.

AR: Leaving aside whether they should or not, then it is a fact that they don't. We have well-established observations of sea-ice deformation at about 10 km resolution, and these are reproduced only by VP models when run at much higher resolution than 10 km. This is because the VP equations assume that the deformation features should be resolved at the grid scale. The brittle rheologies assume that the deformation features result from a sub-grid-scale process (parameterised by the damaging mechanism), and they can therefore reproduce the observed deformation at the observed scale. All of this is too detailed for an abstract, so we will leave the factually correct statement in the paper as it is.

7,8 'independently of resolution'? Why one needs leads and ridges if a mesh is coarse? Do they correspond to any physical reality?

AR: Here, "these features" refers to sea-ice deformation features and not leads and ridges. We have changed the wording accordingly. The mean state in a (relatively) large grid cell will not correspond to a single ridge or lead, but rather to a collection of these, which will influence atmosphere-ocean-ice interactions, as stated in the paper.

11 'give insights ...' This is what I do not really see

AR: We're not sure what to do with this comment. The wording "give insights into" may not be optimal. We have changed this sentence to "We describe the sea ice dynamics and the core of the model in detail and discuss some of the parameters specific to the brittle rheologies included in neXtSIM."

Introduction gives emphasis on historical aspect, and the time intervals of simulations that are mentioned there are discouraging (ten days, an entire year...). I would not not mention this, as it only tells that the model was not really ready.

AR: The point is that the model wasn't ready 10 years ago to do most of what we can do with it today. We wanted to give the reader a sense of how the model has evolved over the last decade and of the effort that went into its development. It also gives a sense of the philosophy behind the model, aiming to reproduce sea ice deformations first. We felt that this was an appropriate framing for the introduction, given that it is a model description paper. We assume that anyone wanting to read a model description paper is already aware of some of the model's uses and is familiar with sea-ice models in general.

64 'on any' ?

AR: Strike any

68 'In it'  What is it?

AR: Replace 'it' with 'the EB model'

74 cascade Marsan

AR: Fixed citation: "... cascade (Marsan … "

95 $P\_max$ is very similar to the sea ice strength in Hibler's rheology, do you select similar parameters as in (16)? Please specify

AR: Equations (6) and (16) (now 35 and 48) have indeed the same form, but the physics behind them differs. In BBM, this is the threshold between elastic and visco-elastic deformation, while in VP, this is the plastic limit. Both are substantial simplifications of the underlying physical processes. We don't want to draw attention to the arguably superficial similarity between the two equations and risk readers conflating them.

119 (P0) and (P1) should be explained. Also it should be explained how derivatives are computed, and there are other things, see

AR: We've added a section detailing our use of the finite element method, so there is no need to mention the P1-P0 discretisation here.

145 Why n+1 in the absolute value $|u\_w-u^{n+1}|$? I hope a linearized version is used.

AR: This was an error in the manuscript. We have corrected it to n

147 'donates'?

AR: This section is substantially rewritten, and this formulation has been removed.

156 stress tensor was boldface initially, please be consistent, there are several places before

AR: We have taken care to use boldface for tensors consistently

168 'particularly after'  It would be better to cite some examples.

AR: We have added a few examples

formula (22) C? it was Ch and Cm before

AR: We were using C as a short-hand for Ch and Cm, but this is not clear enough. We now have two separate equations for Ch and Cm

406 10^5 seems to be rather limiting, a 1/12 degree mesh has already much more cells in the Arctic.

AR: It is limiting. We now also note that this limits Arctic regional setups (the ones the model is most commonly used for) to resolutions coarser than about 5 km.

468 'Some loss ...' Can it be quantified?

AR: We can force the model to remesh at every time step and compare this to a standard run. Note, however, that two standard runs will not give bit-wise the same results, due to random noise added because of uncertainties in model numerics (see e.g. Korosov et al., 2023). The differences caused by random noise are of the same order of magnitude as those caused by not remeshing at every time step. We therefore conclude that this loss of accuracy does not affect the solution in any practical sense, even though we cannot quantify it precisely. The sentence now reads "Some loss of accuracy occurs because the mesh is assumed to remain stationary between remeshing steps, but this has no greater impact on the solution than numerical noise inside the model itself (discussed in Korosov et al., 2023)."

481 'This sampling ...' Why this sentence is needed?

AR: It's not needed, and we have removed it.

527 'Such rapid change ....' See my point 5.

AR: Discussed above

Fig. 2, caption, 100 by 100? (It is 1000 by 1000 in the text).

AR: Fixed

Also add an explanation for b and c.

AR: Done.

Fig. 3, caption, why a,c,e  and b,d,f are grouped together? Please arrange consistently. Also in the text (line 508, a and f?).

AR: We have fixed inconsistencies in labelling of the panels, both in the caption and text.

---

## Author Comment (AC2)

**Response to comments from reviewer #2**

(Reviewer comments are in black, and our responses are in blue)

This paper documents the recent development and state of the neXtSIM sea ice model. The paper is linked with a code release and is a useful documentation of this high quality sea ice model. The model description appears to be complete and is accompanied by some example simulations in stand-alone configurations. These simulations are well described with generally well documented plots.

The paper does need a lot of minor edits though. There are many undefined (or not clearly located definitions) for key terms in the many equations. The equations seem to be all present and correct, but as it is, it is cumbersome to understand them all as I had to a lot of searching around to find the key terms, and in several cases make large assumptions about what is represented.

AR: We have substantially expanded the model description and tried to address those issues in the revised version. The initial submission was meant to be a short, concise description, but it clearly did not work very well.

Another aspect that seems missing are some overview plots from the 15 year run. All the observational comparisons are well documented, but there is limited documentation of the results of the model that emphasise the unique aspect of neXtSIM – the damage based rheology and Lagrangian grid. Can some extra figures be included that show the mean Arctic wide drift fields (that can be compared to observations), mean damage (winter only seems most appropriate) and stress states?

AR: We have added a figure showing the mean Arctic-wide drift field, with a comparison with observations. We have also added a figure and a short section on the modelled damage and stress fields.

Minor edits:

Many equations are missing citations - this is not an issue for the accuracy of the paper, but these documentations are often used for the design of other models. In this case it is essential to be able to search for the derivation of the equations or for more complex forms if this is needed. A table of notation may be a useful addition. While sizable for a paper, it may really help with complexities. It will also make future use or adaptation of these equations much easier.

AR: We have added several references for the equations, hoping that it is now sufficiently detailed. We haven't included a table of notation. A single table of notation is quite large and difficult to order. We are afraid this would ultimately not be very useful. Other options would be just to tabulate the main quantities or to have a separate table for each section. The former option seems insufficient to us, while the latter is overly detailed.

The link between Lagrangian moving mesh and the overall resolution is not clearly defined. The adaptable grid will gain higher or lower resolution as it distorts. It looks like this is all considered as the grid adapts, but is a target resolution (like the 10km for the square example) encoded into this process? Resolution is very important when setting the solver frequency of fixed mesh grids and is generally designed alongside stability and dimensionality. Is there a similar process here?

AR: This is a good point. We have added the following to the second paragraph of the Lagrangian advection and remeshing section.

> The initial mesh sets the mesh resolution, and subsequent adapted meshes retain that same resolution. This is achieved by creating a spatially varying field of mean vertex length for the initial mesh, and then requiring that the lengths of the vertices of later adapted meshes must fall within 20% of the mean vertex length of the initial mesh.

Specific edits:

Abstract,

The opening 5 sentences here are very vague and do not sell this paper well. It needs opening statements on what sea ice is and why dynamical sea ice models exist. Feltham (2008) is a good place to get this narrative and leads well into the intent of neXtSIM to make the link between continuum and Basin scale circulation and the sub grid cell length and observed sea ice deformational features. This is attempted in lines 5-8, but it is all lacking a detailed context.

AR: We have added a paragraph at the start of the introduction to provide better context and motivate the model development. We have also modified the abstract in the same vein.

L 22 – ten simulated days I assume?

AR: Yes. We have specified this.

L 23 these technical terms and the reference to the equation need the equation to be included here. The chosen historical intro is nice, but as it is it needs to have the technical terms removed. A summary of the general improvements and simplified assumptions will help here, with just references to later sections. It may also help to include the definition of Lagrangian, which while a base theory, is an important specification of this particular model.

AR: We have removed the reference to the momentum equation and rephrased parts of this paragraph following the reviewer's suggestions. We also included a very brief explanation of what "Lagrangian advection" means in this context.

– the mix of acronym and citation is awkward to understand at first reading.

AR: We put the acronym in parentheses of its own.

– citation is in the wrong format or missing words.

AR: We put the citations inside parentheses (as they should have been)

– this sentence is unnecessarily long – can it be rewritten with the word 'documented' only once?

AR: We have rewritten it like this: This paper aims to provide an overview of the features of neXtSIM. It will focus on new features, while still giving a relevant description of those parts of the model already documented elsewhere.

'most important' does this mean that it is of the authors opinion that RS obs are more important than in-situ or experimental data? Or that this model design focuses on RS obs matching rather than other obs?

AR: The latter. We have rewritten this sentence to read: A core tenet of the neXtSIM development process is to use the simplest modelling approach that reproduces observations. In this, our main focus has been on satellite remote sensing observations.

'the momentum eq' this suggests that there is only one form of this eqn, the next sentence says that one has been selected for this model – reword. Perhaps say that the core equation conserves momentum?

AR: It's really Newton's second law, but we find that it's generally referred to as the momentum equation in the sea-ice literature. We have changed the text to read: "The core equation of sea-ice dynamics is the momentum equation. This is Newton's second law, but implementations may vary depending on the level of detail considered (e.g. Hibler, 1979; Connolley et al., 2004; Bouillon and Rampal, 2015; Danilov et al., 2015). The form used in neXtSIM is …"

Aph –A is undefined here.

AR: We have added a definition of $A$

– EVP and mEVP are not defined

AR: Added definitions of EVP and mEVP, with references

– stating 'Elasticity E' will make the next eqn better defined.

AR: Added

– '[. ['

AR: Replaced by $0 \leq d < 1$

-  if d is in [0,1] starts at d=0, how can d then be reduced?

AR: That was a mistake; it should have said "increased". We have fixed that now.

– does a local increase in d represent damage to the sea ice? (this may be obvious, but stating the obvious can be helpful early in a paper).

AR: Yes, it does, and we have added this to the paper.

how does this damage reduction occur from a stress point of view? Is it due to the damage and eqn 2 causing stresses to not be transferred across the damaged grid cell?

AR: An increase in damage results in a decrease in elasticity. This means that the ice in the element cannot support as much stress as the neighbouring elements, causing a reorganisation of the stress field over the next time steps. This reorganisation typically results in damage to neighbouring grid cells and a propagation of damage. To try to convey this better, we have rewritten the paragraph in question like this:

> At the start of the simulation, d = 0 everywhere, but d is then increased to ensure that all stresses in the ice are always within the yield criterion. A local increase in d represents damaging of the ice, which is modelled as a reduction in elasticity. The reduction in elasticity in an element means that this element deforms more easily than before, and so the distribution of stresses in the neighbouring elements must change. This causes a stress redistribution and a cascade of damage increases, emulating the multiplicative cascade that Marsan et al. (2004) suggested was the reason for the spatial scaling of sea-ice deformation they observed in satellite remote sensing data.

– observed in RS data or in the model behaviour?

AR: In RS data. We have added this clarification in the manuscript

– I'm not sure what is meant by 'a dashpot'

AR: The sentence was missing an "or": "... introduced a viscous element, or a dashpot, in series …". A dashpot being a common alternative term for "viscous element"

is \eta the viscosity? I don't' think this has been defined yet?

AR: Yes, that was not clear, but is clarified in the revised version.

P is yet to be defined – so thus Pmax is not defined here.

AR: We have added a definition of P

– sigma has a dot in eqn 4, but not in the definition. This suggest one is not meant to be a rate of change.

AR: Yes, it is not the rate of change in equation 7, but rather an intermediate stress value. We have added the following, explaining how we arrive at equation 7 (which is now no 32 in the revised manuscript): "Equation (32) is solved together with the momentum equation using equations (19) and (20). The time stepping of (32) is done in two steps. First, we calculate an intermediate stress, $\boldsymbol{\sigma}'$ through an Euler forwards iteration of \eqref{eq:BBM}, assuming $\dot d = 0$ and $\dot{\boldsymbol\sigma} = (\boldsymbol{\sigma}' - \boldsymbol{\sigma}^n)/\Delta t_m$, which gives"

– see above for P definition.

AR: Done

Eqn5 do the three options for P here relate to the 3 stages in figure1? Can the stages here be directly linked in the text to the parts of the equation? This will help the understanding of this complex rheology.

AR: We now note the relationship between equation 5 and the three stages in figure 1 in the text.

– why is there a scaling about h_0 = 1m? For this value h_0 disappears from eqn 6 which suggests that is useful as a parameter later, otherwise why is it here? (other than dimensionality consistency)

AR: It's there for dimensionality consistency, as you say. The point is to have the units of P not depend on the value of f.

Eqn 7 is the top of this eqn related to the defn of lambda in line 91? If so it is not clear what parameters are put into this equation – what is delta tEK? In particular K? Similar for the bottom line (but this may just be values).

AR: We have rewritten these paragraphs to explain better how equation 7 derives from equation 4. K is now also defined in the paper.

Eqn 8 is this just a solver step, or does this change in d have a differential equation form for the rate of change in damage?

AR: It could have a differential equation form (as per Dansereau et al., 2016, and Ólason et al., 2022), but here it is the factor needed to relax the stresses back onto the yield criterion. We have rearranged the surrounding paragraphs substantially to make this clearer.

Eqn 9 is this the crucial step of the rheology where damage is related to stress? Is this numerically where damage to sea ice is caused by imposed, and thus internal stresses? If so then can the selection of N be explained?

AR: Yes, that is correct. We have rewritten the paragraphs around this equation to make this clearer. The selection of N is also explained and quantified: "A capping with a large value of N is needed to prevent numerical instabilities (see Plante et al., 2020), where N is chosen to be sufficiently large to avoid impacting the solution. In neXtSIM, the default value is four orders of magnitude larger than the cohesion."

is the conceptual form of P the same for both rheologies?

AR: Equations (6) and (16) (now 35 and 48) have indeed the same form, but the physics behind them differs. In BBM, this is the threshold between elastic and visco-elastic deformation, while in V,P this is the plastic limit. Both are substantial simplifications of the underlying physical processes. We don't want to draw attention to the arguably superficial similarity between the two equations and risk readers conflating them.

Eqn 17 – this split line form has gone wrong somehow

AR: Yes, it's formatted for the eventual two-column format, so it will look nicer when published.

- can all the elements of F be seen in eqn 1?

AR: We now write this equation out in full, not using F (equation 52)

Eqn 18 – how does beta' in line 146 relate to beta here and on line 153?

AR: \beta' was defined following equation (17) - line 146. In the revised version, it now says "\beta' is is the numerical tuning parameter introduced (as just \beta) by Bouillon et al. (2013)"

Hunke citation and text is confusing – the wrong way around perhaps?

AR: Yes, we've fixed this.

– it is not clear if the beginning of this paragraph is for ice models in general or specific to NextSim

AR: It refers to sea-ice models in general. We've made this clearer by prefacing the first sentence with "In general, …"

Eqn 19 -  this look similar to the Lepperanta 1993 review where the thermos model for sea ice is derived. Is this the source for this version and does it have a similar derivation from the original heat equations?

AR: It's a simplification of Semtner's (1976) equations. You are right that Lepperanta does something similar, but it's not directly the source here. We've added a reference to Semptner's paper.

-  for the two layer model – how is the lower layer solved?

AR: As Winton's 2000 paper is well known, we don't consider it relevant to go into the details here. Rather, we have added the following to the end of the paragraph in question: "Once we have calculated the surface fluxes, we calculate ice and snow thickness changes and internal ice temperatures for the two-layer model, following Semtner (1976) for the zero- layer model and Winton (2000) for the two-layer model."

Eqns 20-21 these need citations.

AR: These equations are from Stull (1988), p262, equations 7.4.1d and 7.4.1e [Stull, Roland B. An introduction to boundary layer meteorology. Vol. 13. Springer Science & Business Media, 1988.]

-  the difference between C, Ch and Cm is not defined – I can't find where Cm is used up to this point. Eqn 22 does not have a way to differentiate between Cm and Ch. There is also Cb in eqn 1.

AR: We have now replaced equation 22 with two equations, one for C_h and one for C_m

Eqn 23 k and g undefined here

AR: k is the von Kármán constant, as noted after the preceding equation, and g is the gravitational acceleration (this is now noted).

– it looks like Cm,h develop in time from this statement, but the previous eqns do not make it clear why and how this happen – what is the reason why they are not just calculated at each time step and therefore need to develop?

AR We now use the n and n+1 superscripts for C_h and C_m to make this clear. The last paragraph is also expanded to read:

As advecting additional prognostic variables is very cheap, we set the drag coefficients as prognostic variables and can calculate $C_h^{n+1}$ and $C_m^{n+1}$ from $C_h^n$ and $C_m^n$. In models where advection is more expensive, a common approach is to solve for drag iteratively, starting from the neutral drag and recalculating it about 5 times to obtain a more accurate estimate of the stability-dependent drag. The atmospheric stability changes slowly enough that using the drag coefficient from the previous time step is sufficient to calculate the Obukov length and the drag coefficient for the current time step.

is this figure 9 and 10 in Webster et al or here?

AR: In Webster et al. This is now made clear.

are there citations for where these assumptions come from or are they new?

AR: They are new. This is now made clear in the text.

– is level-ice a separate tracer class, or just a theoretical area for the conservation law? – edit -  can this section begin with which of the following S,H,R are recorded by the model?

AR: "Level-ice" is a theoretical construct here. We have rearranged and rewritten this section following the suggestions in this and the following comment, following the reviewer's suggestions.

while the above list and equations are both clearly written can the prior points be referred to here to improve clarity? Or the whole section can be formatted into a list so the equations are linked to the prior numbered laws? As it is the list is repeated again for each equation.

AR: See above.

where do such open boundaries occur? Later text suggests it is the sea ice edge, but perhaps for a reduced domain?

AR: Here, open boundaries refers to the open boundaries of a reduced domain. This is now clarified in the text.

From a previous point – the recorded tracers are defined here – a reference to this point in the earlier sections will help.

AR: This is now noted can the adaptation mentioned here refer back to previous list where it is first mentioned?

AR: We have rewritten the text so that it always refers back to the list

Algorithm 1 – while this is complete and the description accurate- it will be really help the reader if the separate blocks of the pseudo-code can be referred to in the text to link the description to the relevant parts. Perhaps line numbers can be added here and referred to in the text?

AR: We have implemented this suggestion, using line numbers in the algorithm and referring back to those in the code in this sentence it is unclear whether this describes two outputs, or a single one with two forms of data, can numbers be added to split them? (assuming it describes two outputs – I think it does?)

AR: We have rephrased this sentence to read "NeXtSIM supports two output formats: snapshots of the model state with complete mesh information on the one hand, and netCDF output on a fixed, rectangular output grid on the other."

– can a physical or rheology description of the difference between these two experiments be added here (it Is touched upon later)? This will make the following descriptions have stronger context.

AR: We have added a paragraph explaining better the difference between the two damage relaxation schemes used.

figure two shows deflection of the ice drift -  this suggests a coriolis acceleration, is this true? What value/global location is used here?

AR: Yes, the deflection is due to the Coriolis acceleration. The domain is located with the geographic North Pole at its lower left corner. We now mention this in the first paragraph of this sub-section.

– does this refer to the green line in 3c?

AR: Yes, this is now noted in the text

Figure2 – the caption needs to show what time point these figures represent (it is in the text but needs to be here too). similar point for figure 3.

AR: Done

-  why is higher damage localisation better?

AR: Changed to "higher damage localisation", which was the intended meaning.

this next sentence suggests that small scale deformation lines are a problem?

AR: They are not; this was unfortunate phrasing. We moved the mention of the small-scale deformation lines to the end of the previous sentence.

5.2 -  tuning is mentioned later when discussing the results. What tuning was performed prior to this model run, or was the experiment repeated at all with alternate parameters?

AR: This particular experiment was not repeated with alternate parameters, but we have experience tuning the model to increase the consistency with observations. The tuning of a model depends on the scientific question at hand; it would be difficult to provide a quick overview of what is possible here. Instead, we refer the reader to studies in which the tuning of neXtSIM has been discussed. We added references to Ólason et al., 2022; Boutin et al., 2023; and Korosov et al., 2025.

Figure 5 and near line 575. Can you comment on the think ice near the Siberian coast that is not seen in the observations? Is this related to drift at all? Why are no maps of mean drift speed presented ?

AR: This thick ice was the result of an error in the building of the climatology, resulting in only 1 year being taken into account, and this thick ice was likely the result of ridging near the landfast ice for that particular year. Re-doing the climatology (now Figure 7), this thick ice feature has disappeared (note that we also reduced the colourmap amplitude). We think commenting in detail on the differences between the model and observations, and discussing the processes, is out of scope here; we only intend to showcase the mean state of the sea ice simulated by neXtSIM. The aim is to give an idea of what to expect when running the model.

We have also added a figure presenting maps of the mean drift, as recommended by the reviewer.

Feltham, Daniel L. 'Sea Ice Rheology'. Annual Review of Fluid Mechanics 40, no. 1 (January 2008): 91–112. https://doi.org/10.1146/annurev.fluid.40.111406.102151.

Leppäranta, Matti. 'A Review of Analytical Models of Sea-ice Growth'. Atmosphere-Ocean 31, no. 1 (1 March 1993): 123–38. https://doi.org/10.1080/07055900.1993.9649465.